# CHOOSING PUBLIC DATASETS FOR PRIVATE MACHINE LEARNING VIA GRADIENT SUBSPACE DISTANCE

## ABSTRACT

Differentially private stochastic gradient descent privatizes model training by injecting noise into each iteration, where the noise magnitude increases with the number of model parameters. Recent works suggest that we can reduce the noise by leveraging public data for private machine learning, by projecting gradients onto a subspace prescribed by the public data. However, given a choice of public datasets, it is unclear why certain datasets perform better than others for a particular private task, or how to identify the best one. We provide a simple metric which measures a low-dimensional subspace distance between gradients of the public and private examples. We empirically demonstrate that it is well-correlated with resulting model utility when using the public and private dataset pair (i.e., trained model accuracy is monotone in the distance), and thus can be used to select an appropriate public dataset. We provide theoretical analysis demonstrating that the excess risk scales with this subspace distance. This distance is easy to compute and robust to modifications in the setting.

## 1 INTRODUCTION

Recent work has shown that machine learning (ML) models tend to memorize components of their training data (Fredrikson et al., 2015), and in fact attackers can often recover training samples from published models through carefully designed attacks (Carlini et al., 2021; Shokri et al., 2017). This is a critical privacy issue when models are trained on private data. A popular approach to address this issue is to adopt *Differential Privacy* (DP) (Dwork et al., 2006) as a rigorous privacy criterion that provably limits the amount of information attackers can infer about any single training point. *Differentially private stochastic gradient descent* (DPSGD) (Abadi et al., 2016; Song et al., 2013; Bassily et al., 2014) is one of the most commonly used methods to train a ML model (differentially) privately. It makes two main modifications to vanilla SGD: 1) clipping per-sample gradients to ensure a bound on their $\ell_2$ norms; 2) adding Gaussian noise to the gradient.

One downside of adopting DP in ML is that we need to sacrifice utility of the trained model to guarantee privacy. Specifically, DPSGD noises the gradient at each step, with noise drawn from a spherical Gaussian distribution, $\mathcal{N}\left(\mathbf{0}, \sigma^2 \mathbb{I}_{p \times p}\right)$, where $p$ is the model dimension (i.e., the number of model parameters) and the variance $\sigma^2$ scales the noise. In order to bound the privacy leakage, the magnitude of noise introduced in each step must scale with the square root of the number of parameters $p$. Consequently, for many large models, the noise introduced may overwhelm the signal contributed by the original gradients, significantly diminishing the utility.

Several works have proposed methods to improve the utility of private machine learning (Zhou et al., 2021; Golatkar et al., 2022; Luo et al., 2021; Yu et al., 2021; Kairouz et al., 2021; Nasr et al., 2023; Ganesh et al., 2023). One fruitful direction uses *public* data, i.e., data that is not subject to any privacy constraint. There are primarily two types of approaches that incorporate public data in private training. The first involves *transfer learning*, where we pretrain the model on a public dataset and then (privately) finetune the model on a sensitive dataset for our target task (Luo et al., 2021; Abadi et al., 2016; Yu et al., 2022; Li et al., 2022c). Another approach is based on *pre-conditioning*, which exploits the empirical observation that during training, the stochastic gradients (approximately) stay in a lower-dimensional subspace of the $p$-dimensional gradient space. Consequently, some works find this subspace using the public data, and then project the sensitive gradients to this (public) subspace before privatization (Zhou et al., 2021; Golatkar et al., 2022; Yu et al., 2021; Kairouz et al., 2021).

This reduces the magnitude of the introduced noise and generally improves utility over DPSGD without supplementary data.

However, this raises a number of natural questions. From a scientific perspective: why do some public datasets work better than others for a particular private task? One may intuitively believe that public datasets which "look relevant" to the private task would perform best, but this is not a precise statement, and furthermore (as we will empirically demonstrate) may be misleading or incorrect. Building on this question, from a pragmatic standpoint, how should one select which public dataset to use?

Our main contribution addresses both of these questions: we provide a simple metric for measuring the distance between datasets, and show that it is very well-correlated with model utility when treating one dataset as private and the other as public.

Table 1: GEP evaluation AUC and corresponding distance in descending order. We use the *same* model setting for private training and distance computation. "-" means DP-SGD training without using any public data. Mean and standard deviation are calculated over 3 runs.

| AUC | Private Dataset | Public Dataset | Distance |
|---|---|---|---|
| **68.93% / 0.05** | | ChestX-ray14 | **0.15** |
| 67.22% / 0.21 | | SprXRay | 0.32 |
| 67.20% / 0.34 | | CheXpert | 0.32 |
| 66.61% / 0.04 | ChestX-ray14 | KagChest | 0.36 |
| 65.37% / 0.43 | | Kneeos | 0.38 |
| 64.76% / 0.16 | | - | - |
| 48.60% / 0.02 | | CIFAR-100 | 0.55 |
| Time cost for each distance computation: 12s | | | |

We demonstrate its efficacy in both transfer learning and pre-conditioning settings. To summarize our contributions:

1. **We introduce Gradient Subspace Distance (GSD), a metric to quantify the difference between private and public datasets.** GSD is an easily computable quantity that measures the distance between two datasets.

2. **We find GSD is well-correlated with model utility when selecting public datasets in both pre-conditioning and transfer learning settings.** As a representative example, Table 1 shows the utility of a privately trained model using a public dataset increases monotonically as GSD decreases. Our theoretical analysis demonstrates that the excess risk of Gradient Embedding Perturbation (GEP) (a private training algorithm that leverages public data for gradient pre-conditioning) scales with the GSD.

3. **We show that GSD is *transferable*.** The ordering of GSD for several choices of public dataset remains fixed across architectures, both simple (e.g., 2-layer CNN) and complex. Using these simple architectures as a proxy, we can efficiently compute GSDs which are still useful for privately training large models.

## 2 RELATED WORK

**Transfer Learning.** In the differentially private setting, it is now common to pre-train a model on public data, and then privately fine-tune on private data. This can result in comparable utility as in the non-private setting, evidenced for both language models (Yu et al., 2021; 2022; Li et al., 2022c) and vision tasks (Luo et al., 2021; De et al., 2022; Mehta et al., 2022). In many cases, due to computational requirements, it may be challenging to pre-train a large model on a public dataset. Instead, many practitioners will turn to pre-trained weights, which obviate the computational burden, but give less flexibility to choose an appropriate training dataset. As a result, we use second-phase pre-training, in which we perform a second phase of pre-training with a modestly-sized public dataset. This has been proven to be useful in non-private setting (Gururangan et al., 2020).

**Pre-conditioning.** Empirical evidence and theoretical analysis indicate that while training deep learning models, gradients tend to live in a lower-dimensional subspace (Gur-Ari et al., 2018; Li et al.,

2020; 2022b; Golatkar et al., 2022; Kairouz et al., 2020). This has led to methods for private ML which project the sensitive gradients onto a subspace estimated from the public gradients. By using a small amount of i.i.d. public data, Zhou et al. (2021) demonstrate that this approach can improve the accuracy of differentially private stochastic gradient descent in high-privacy regimes and achieve a dimension-independent error rate. Similarly, Yu et al. (2021) proposed GEP, a method that utilizes public data to identify the most useful information carried by gradients, and then splits and clips them separately. Amid et al. (2022) proposed a DP variant of mirror descent that leverages public data to implicitly learn the geometry of the private gradients. In the first-order approximation, it can be considered as using public gradients as a regularizer for DP-SGD. Gradient estimation based on public data can also be used as a preconditioner for adaptive optimizers like RMS-Prop (Li et al., 2022a).

**Domain Adaptation.**   We aim to quantify the similarity between private and public datasets. One related area of research is distribution shift, or domain adaptation (Gabrel et al., 2014; Wang & Deng, 2018; Zhuang et al., 2019; Zhang, 2019; Ben-David et al., 2010; 2006; Wang et al., 2021). At a high level, research in this area examines the problem of when the distributions of test and training data differ, which aligns with our goals. However, most work in this area focuses on reducing the gap between in- and out-of-distribution test errors, where target data is used repeatedly for accuracy improvement. Most of the work along this line assumes that the target data is also public or doesn't consider privacy, and is thus inappropriate for the private learning setting. To the best of our knowledge, the only work with a similar focus to us is Task2Vec (Achille et al., 2019), which uses the Fisher information matrix to represent a dataset as a vector, allowing for the measurement of a distance between two datasets. However, it is not suitable for private learning tasks as our empirical evaluation shows that Task2Vec fails to accurately rank the utility of public datasets.

**Choosing Proper Public Data For Private ML.**   Perhaps the most comparable work is the independent and concurrent work of Yu et al. (2023), as both aim to select the best public data for private machine learning. Yu et al. (2023) propose a private learning framework that uses private dataset selection to choose a subset of the pre-training dataset. While our goals are similar, there are differences in our methods and use cases. Our work focuses on scenarios with limited amounts of public data. In contrast, Yu et al. (2023) focuses on first-phase pretraining with large-scale datasets (see Figure 2). However, this is only applicable for organizations with large computational resources, who can afford to pretrain such large models. In contrast, we believe most practitioners and researchers (ourselves included) are much more compute-constrained, and will rely upon already pre-trained models, with little control over the first-phase pretraining set. This is the focus of our work. Finally, their method requires training a full-sized DP model, whereas our method can be executed in less than a minute.

## 3   PRELIMINARIES

**Notation.**   We use $p$ to denote the model dimension, i.e., the number of parameters in the model. $k$ is a parameter we will use to denote the dimension of the lower-dimensional space we choose. $m$ refers to the number of examples in a batch. We use superscripts and subscripts interchangeably to denote private or public data, like $x_{priv}$, $V^{pub}$.

**Definition 1 (Differential Privacy (Dwork et al., 2006)).**   *A randomized algorithm $\mathcal{A}$ is $(\epsilon, \delta)$-differential private if for any pair of datasets D, D' that differ in exactly one data point and for all subsets E of outputs, we have:*

$$\Pr[\mathcal{A}(D) \in E] \leq e^{\epsilon} \Pr[\mathcal{A}(D') \in E] + \delta.$$

**Definition 2 (Projection Metric (Ham & Lee, 2008; Edelman et al., 1998)).**   *The projection metric between two $k$-dimensional subspaces $V_1$, $V_2$ is defined as:*

$$d\left(V_1, V_2\right) = \left(\sum_{i=1}^{k} \sin^2 \theta_i\right)^{1/2} = \left(k - \sum_{i=1}^{k} \cos^2 \theta_i\right)^{1/2}$$

*where the $\theta_i$'s are the principal angles (see Appendix A) between $V_1$ and $V_2$.*

**Gradient Embedding Perturbation (GEP).**   Our theoretical analysis is based on GEP (Yu et al., 2021), a private learning algorithm that leverages public data for gradient preconditioning. We note

that GEP is the leading approach among private ML algorithms that use public data for preconditioning, and produces state-of-the-art results. Hence, for preconditioning-based methods, we focus entirely on GEP. Here we briefly introduce their algorithm. GEP involves three steps: 1) it computes a set of the orthonormal basis for the lower-dimensional subspace; 2) GEP projects the private gradients to the subspace derived from step 1, thus dividing the private gradients into two parts: embedding gradients that contain most of the information carried by the gradient, and the remainder are called residual gradients; 3) GEP clips two parts of the gradients separately and perturbs them to achieve differential privacy. The full algorithm is in Appendix A.

## 4 Gradient Subspace Distance

Suppose we have a task that consists of a private dataset $X^{priv}$ and a differentially private learning algorithm $A$ that can leverage public data to improve model utility. We have a collection of potential choices of public dataset $[X_1^{pub}, X_2^{pub}, \cdots]$. We would like a metric that, when computed for a public dataset $X^{pub}$, is well-correlated with utility when $X^{pub}$ is used as the public dataset with algorithm $A$ on private task $X^{priv}$. This serves two purposes: first, it gives a quantitative metric that can formalize dataset similarity and suitability for use as a public dataset. Second, it may be useful in actually *selecting* a public dataset for use for a particular private task, which is a critical hyperparameter.

---

**Algorithm 1** Gradient Subspace Distance (GSD)

---

**Input:** Private examples $x_{priv}$, public examples $x_{pub}$, loss function $\mathcal{L}$, model weights $\mathbf{w}_0$, dimension $k$

**Output:** Distance between two image datasets $d$

1: // Compute per-sample gradient matrix for private and public examples
2: $G_{priv} = \nabla \mathcal{L}(\mathbf{w}_0, x_{priv})$
3: $G_{pub} = \nabla \mathcal{L}(\mathbf{w}_0, x_{pub})$
4: // Compute top-$k$ subspace of the gradient matrix
5: $U^{priv}, S^{priv}, V^{priv} \leftarrow \mathbf{SVD}(G_{priv})$
6: $U^{pub}, S^{pub}, V^{pub} \leftarrow \mathbf{SVD}(G_{pub})$
7: // Compute the distance between two subspaces
8: $d = \mathbf{ProjectionMetric}(V_k^{priv}, V_k^{pub})$

---

We present the pseudo-code of our algorithm, Gradient Subspace Distance (GSD) in Algorithm 1. At a high level, our method involves the following two steps: finding the gradient subspace of the public and private data examples, and computing their gradient subspace distance. The algorithm uses the same model $A$ and a batch of randomly labeled data examples from private and public datasets. Following standard DPSGD, the algorithm will first compute and store per-example gradients from each data example, that is $G_{priv}, G_{pub} \in \mathbb{R}^{m \times p}$. Then it computes the top-$k$ singular vectors of both the private and public gradient matrix by performing singular value decomposition (SVD). Finally we use projection metric to derive the subspace distance $d$ by taking the right singular vectors $V_k^{pub}, V_k^{priv}$ from the previous step.

GSD is naturally suited to the aforementioned pre-conditioning methods. In each iteration, these methods project the private gradients to a low-dimensional subspace, which ideally contains most of the signal of the gradients (In Appendix D.2, we empirically reconfirm that using the top subspace of the gradients themselves contains most of their signal). Since repeatedly selecting the top subspace of the gradients themselves is not a privacy-preserving operation, we instead choose a public dataset to use as a proxy. Thus intuitively, a public dataset with a "similar top subspace" should be suitable. This is what GSD tries to capture, and the best dataset should be the one with minimum GSD.

However, following this intuition only gets us so far: taking it literally would measure distances between the public and private datasets at each step throughout the training process, an impractical procedure that would introduce significant overhead. Remarkably, we instead find that a simple alternative is effective: compute the distance only once at initialization (Section 4.2). This requires only a single minibatch of each dataset, and as we show in our experiments, is surprisingly robust to changes in model architecture (Section 6.3). Most importantly, we show that it is also effective for *transfer learning* settings (Section 6.2), where subspace projections are not used at all, thus

demonstrating that GSD more generally captures dataset similarity and fitness-for-use of public datasets.

## 4.1 EXCESS RISK SCALES WITH GSD

In this section, we theoretically prove that the excess risk of a pre-conditioning public-data-assist private machine learning algorithm, e.g., GEP (Yu et al., 2021), is bounded by Gradient Subspace Distance (GSD) under standard statistical learning assumptions. We first show that the reconstruction error $\|\mathbf{G}_{priv} - \mathbf{G}_{priv}V_k^{pub}V_k^{pub\top}\|_2$ is bounded by GSD. Then we show that the convergence bound of excess risk is determined by the reconstruction error.

**Lemma 4.1.** *For GEP, let $\mathbf{G}_{priv}$, $V_k^{pub}$, $V_k^{priv}$ be the gradient matrix and top-k gradient subspace from public or private examples at step t, respectively. Then we have the spectral norm of reconstruction error*

$$\|\mathbf{R}\|_2 \leq \sqrt{2}s_{1,t}\mathbf{GSD}(V_k^{priv}, V_k^{pub}) + s_{k+1,t} \tag{1}$$

*where $\mathbf{R} = \mathbf{G}_{priv} - \mathbf{G}_{priv}V_k^{pub}V_k^{pub\top}$ is the reconstruction error of private gradient matrix $\mathbf{G}_{priv}$ using public examples, $s_{1,t} \geq ... \geq s_{k,t} \geq ...$ are the singular values of $\mathbf{G}_{priv}$, $\mathbf{GSD}(V_k^{priv}, V_k^{pub})$ is the gradient subspace distance given by our algorithm.*

Lemma 4.1 indicates that the reconstruction error of the private gradient matrix using public examples at step $t$ is bounded by GSD, the subspace distance between the public and private gradient subspaces. A larger GSD may yield a larger reconstruction error at each step. Proof of this lemma can be found in Appendix E.

**Theorem 4.2.** *Assume that the loss $L(\mathbf{w}) = \frac{1}{n}\sum_{i=1}^n \ell(\mathbf{w}, z_i)$ is 1-Lipschitz, convex, and $\beta$-smooth. Let $\mathbf{w}^* = \operatorname{argmin}_{w \in \mathcal{W}}L(\mathbf{w})$. The excess risk of GEP obeys*

$$\mathbb{E}[L(\overline{\mathbf{w}})] - L(\mathbf{w}^*) \leq O\left(\frac{\sqrt{k\log(1/\delta)}}{n\epsilon}\right) + O\left(\frac{\sqrt{p\log(1/\delta)}}{n\epsilon}\overline{\mathbf{d}}\right) \tag{2}$$

*where GEP is $(\epsilon, \delta)$-DP (see Appendix A). Here we set $\eta = \frac{1}{\beta}, T = \frac{n\beta\epsilon}{\sqrt{p}}, \overline{\mathbf{w}} = \frac{1}{T}\sum_{t=1}^T \mathbf{w}_t,$ $\overline{\mathbf{d}} = \frac{1}{T}\sum_{t=1}^T d_t^2, d_t = \sqrt{2}s_{1,t}\mathbf{GSD} + s_{k+1,t}$, and **GSD**, s are the gradient subspace distance and singular values of the gradient matrix at step t.*

Theorem 4.2 shows that the excess risk is affected by the GSD at each step. A larger GSD will result in a larger excess risk, which is often evaluated by the error rate in the experiments. Proof of this theorem can be found in Appendix E.

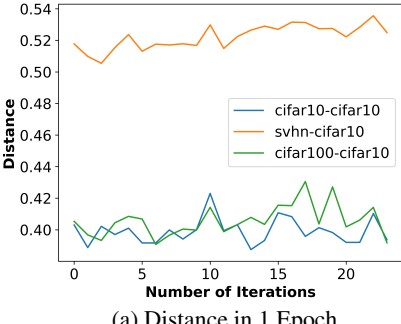 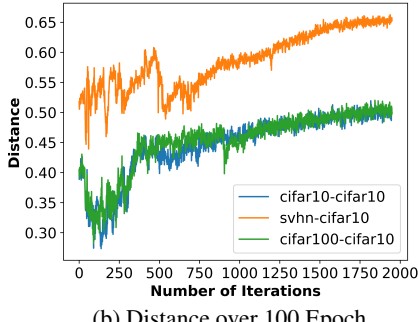

(a) Distance in 1 Epoch        (b) Distance over 100 Epoch

Figure 1: The trend of distance during the process of training a ResNet20 model on CIFAR-10 using vanilla SGD. We follow a standard SGD training procedure and compute the distance between the current private batch and public examples at each iteration.

## 4.2 ORDERING OF GSD IS PRESERVED OVER TRAINING

Theorem 4.2 shows that the excess risk, measured by the error on the test set, can be predicted by GSD, assuming that we have *fresh* private examples at each step. However, this will cause significant

privacy leakage and computational overhead if we repeatedly compute this distance using the whole private dataset.

We empirically measure the GSD for each of the public datasets throughout the training process, as shown in Figure 1. This demonstrates that the relative ordering of the distances is preserved at almost all times. As a result, it suffices to compute each of the GSDs only once at initialization, requiring only one batch of examples from the private dataset and one pass through the model, incurring minimal data exposure and computational overhead. We can then select the dataset with the smallest GSD for use as our public dataset.

## 5 SECOND-PHASE PRE-TRAINING

The standard method of private transfer learning consists of two phases: pre-training on a public dataset and fine-tuning on a private task. However, with large training sets and models, the computational burden of pre-training is prohibitive for most practitioners. Consequently, it is common to instead use pre-trained weights (obtained through pre-training on a fixed dataset) rather than run pre-training on a public dataset of choice. While this is computationally convenient, it limits the choice of pre-training datasets, and thus limits the accuracy in downstream fine-tuning.

$$\underbrace{f(\mathbf{W}_0; x) \xrightarrow{\mathbf{X}^{pub,1}} f(\mathbf{W}_{PT}; x)}_{\texttt{Load pre-trained weights}} \underbrace{\xrightarrow[\theta]{X_{pub,2}} f(\mathbf{W}_{PT}, \theta_0; x)}_{\texttt{Second-phase pre-training}} \underbrace{\xrightarrow{X_{priv}} f(\mathbf{W}_{PT}, \hat{\theta}; x)}_{\texttt{Private fine-tuning}}$$

Figure 2: Second-phase pre-training pipeline. We first choose a large model and download its pre-trained weights. Then we use an appropriate parameter-efficient fine-tuning mechanism and get trainable parameters $\theta$. We train $\theta$ on a public dataset and get $\theta_0$, called ***second-phase pre-training***. Finally, we pass $\theta_0$ as the initial weights for private fine-tuning on the target task.

To alleviate this issue, we consider *second-phase pre-training*, in which a set of pre-trained weights are pre-trained on a second public dataset. We can then (privately) fine-tune the model on a sensitive dataset for the downstream task of interest. While this paradigm has previously been considered in the non-private setting Gururangan et al. (2020), to the best of our knowledge, we are the first to explore second phase pre-training in the differentially private setting. Pre-trained models may be significantly out of distribution with respect to the downstream task. Due to the noise introduced, the ability to adapt during fine-tuning may be diminished under differential privacy. Thus, the additional public data may be valuable for reducing the distribution shift. Second-phase pre-training is illustrated in Figure 2.

### 5.1 SECOND-PHASE PRE-TRAINING STEP BY STEP

Now we formally define second-phase pre-training. Suppose $f(\mathbf{W}_{pt}; x)$ where $\mathbf{W}_{pt}$ denotes pre-trained weights and $x$ is input. To do second-phase pre-training, we first use a parameter-efficient fine-tuning mechanism and create new trainable parameters. Then we train these parameters on some public datasets. This step can be described by:

$$f_{2pt}\left(\mathbf{W}_{pt}, \theta; x_{pub}\right) \rightarrow \theta_0 \tag{3}$$

where $x_{pub}$ are the public datasets and $\theta$ are the new trainable parameters, which are of far lower dimensionality than $\mathbf{W}$. We get the parameter vector $\theta_0$ after this second-phase pre-training step. Finally, we initialize $\theta = \theta_0$ and privately fine-tune it by running DPSGD on the private task:

$$f_{ft}\left(\mathbf{W}_{pt}, \theta_0; x_{priv}\right) \rightarrow \hat{\theta} \tag{4}$$

Our experiments show that second-phase pre-training can give additional accuracy improvements, even when we only have a small number of public data examples. Furthermore, our distance measurement GSD remains a good indicator for choosing good public data for the second phase pre-training.

### 5.2 PARAMETER EFFICIENCY IN PRIVATE FINE-TUNING

In both private and non-private settings, approaches frequently depart from the default of fine-tuning all model weights. For example, one can freeze parameters and fine-tune only specific layers, or

introduce new parameters entirely. The resulting number of tunable parameters is almost always chosen to be smaller than during pre-training, leading to *parameter efficient* methods. This can be beneficial in terms of portability and resource requirements, and the fine-tuned model utility frequently matches or compares favorably to full fine-tuning. Parameter efficiency may be further advantageous in the differentially private setting, as it reduces the magnitude of noise one must introduce (though findings on the downstream impact on utility remain inconclusive). In the settings we consider, we will empirically find that parameter-efficient methods result in better utility.

In general, there are two ways of parameter-efficient fine-tuning. One approach is to select a subset of layers or parameters for fine-tuning. For instance, Bu et al. (2022) proposed fine-tuning only the bias terms of a model, which is both computationally and parameter-efficient while retaining similar accuracy compared to other methods. Another study by Cattan et al. (2022) found that fine-tuning the first and last layers of a model consistently improves its accuracy. The other approach is to freeze all existing parameters and add new trainable parameters during fine-tuning. Some examples include Adapter (Houlsby et al., 2019), Compacter (Mahabadi et al., 2021) and LoRA (Hu et al., 2022). Yu et al. (2022); Li et al. (2022c) demonstrated that private fine-tuning using parameter-efficient methods on large language models can be both computationally efficient and accurate.

## 6 EXPERIMENTS

We explore the predictive power of GSD in both pre-conditioning and transfer learning settings. Specifically, we use GSD to choose a public dataset for GEP (Yu et al., 2021) and DP Mirror Descent (Amid et al., 2022)(representative of pre-conditioning methods) and second-phase pre-training (representative of transfer learning settings). We use a variety of datasets, including Fashion MNIST (Xiao et al., 2017), SVHN (Netzer et al., 2011), and CIFAR-10 (Krizhevsky et al., 2009), as three canonical vision tasks. Based on the recommendations of Tramèr et al. (2022), we also evaluate our methods on datasets closer to privacy-sensitive applications. In particular, we also work with two medical image dataset: ChestX-ray14 (Wang et al., 2017) and HAM10000 (Tschandl, 2018). A variety of datasets are chosen as public data respectively. We evaluate our algorithms using both CNN-based (e.g., ResNet152 (He et al., 2016), DenseNet121 (Huang et al., 2017)) and Transformer-based (ViTs (Dosovitskiy et al., 2020)) architectures. A variety of parameter-efficient fine-tuning mechanisms are considered, including freezing layers and LoRA (Hu et al., 2022). Further details on our experimental setup appear in Appendix B.

We compute GSD non-privately using Algorithm 1, for two reasons. First, as discussed in Section 4, the privacy leakage due to hyperparameter selection is considered to be minimal and often disregarded in private ML. We thus treat selection via GSD similarly. Indeed, we experimentally validate that GSD has minimal impact on privacy using membership inference attacks in Section D.1. Second, beyond being a tool for public dataset selection, it is interesting in its own right to understand which metrics determine dataset utility in transfer settings.

Ideally, we would like our distance measure GSD to be model agnostic: it should depend only the two datasets, not on any particular model. This is not the case, since, as stated, our algorithms take gradients of the two datasets on the model of interest. However, we show that GSD is highly robust to changes in model architecture. We evaluate GSD on a 2-layer CNN (which we call a "probe network"), and show that relative ordering of GSDs is preserved, even though the architecture is far simpler than the models of interest.

We also compare our algorithm with Task2Vec (Achille et al., 2019), which has a similar goal as GSD. At a high level, Task2Vec represents a task (i.e., dataset) by transforming it into a vector so that the similarity between different datasets can be prescribed by the distance between two vectors. Although experiments show that Task2Vec matches taxonomic relations for datasets like iNaturalist (Horn et al., 2018), our empirical evaluation shows that it is outperformed by GSD in the differentially private setting.

### 6.1 RESULTS FOR PRE-CONDITIONING

We compute GSD and evaluate using GEP for the chosen datasets. The evaluation results are in Table 2. We find that, across a very wide range of different (and non-cherry-picked) private and public datasets, final accuracy is monotone as GSD decreases. Unexpectedly, we find that GSD between CIFAR-10 and CIFAR-100 is less than between CIFAR-10 and CIFAR-10. Nonetheless, this is predictive of final performance, where we see using CIFAR-100 as a public dataset is better

than CIFAR-10, despite the fact that the private dataset is also CIFAR-10. In particular, this supports our use of a rigorous metric to predict dataset suitability rather than going off intuition (which this example shows can be incorrect).

Table 2: GEP evaluation accuracy and corresponding distance in descending order. We use the *same* model for private training and GSD computation. "-" means DP-SGD without public data.

| Accuracy | Private Dataset | Public Dataset | Distance |
|---|---|---|---|
| **58.63%** | | CIFAR-100 | **0.20** |
| 57.64% | CIFAR-10 | CIFAR-10 | 0.24 |
| 56.75% | | SVHN | 0.28 |
| 52.16% | | - | - |
| **91.32%** | | SVHN | **0.25** |
| 89.29% | SVHN | CIFAR-100 | 0.31 |
| 89.08% | | MNIST-M | 0.39 |
| 83.21% | | - | - |
| **85.25%** | | FMNIST | **0.34** |
| 84.54% | FMNIST | FLOWER | 0.43 |
| 83.91% | | MNIST | 0.50 |
| 79.77% | | - | - |

For ChestX-ray14, we use AUC instead of prediction accuracy because of high class imbalance. The evaluation results are given in Table 1. Once again, lower GSD implies higher model utility. We see that ChestX-ray14 is the best public dataset, but the second best is another chest x-ray dataset. Furthermore, using a significantly different dataset (CIFAR-100) as the public dataset results in worse utility than using no public dataset at all.

We include further experiments in the appendix: 1) evaluation on DP Mirror Descent, in Appendix D.3; and 2) combining multiple public datasets, in Appendix D.7.

## 6.2 RESULTS FOR SECOND-PHASE PRE-TRAINING

We compute the GSD and evaluate using second-phase pre-training for the chosen datasets. The evaluation results are given in Table 3. As before, we consistently find that smaller GSD leads to larger utility. Like ChestX-ray14, HAM10000 is highly imbalanced, so we again use AUC. However, unlike ChestX-ray14, which contains roughly 100,000 images, HAM10000 is relatively small (only 10,000 skin lesion images). We assume that we can only collect 300 images from it and treat them as public. As shown in Table 3, even this small public dataset can boost the utility through second-phase pre-training. While even the worst public dataset does not dramatically hurt utility (in contrast to the pre-conditioning setting), GSD can still be a good indicator of the utility of public datasets. Similar results apply when we evaluate second-phase pre-training and GSD on ChestX-ray14 using ViTs, as shown in Appendix D.4.

Table 3: Second-Phase evaluation results and corresponding distance in descending order. We use DenseNet121 and choose two convolutional layers and the last layer for second-phase pre-training and private fine-tuning. Detailed settings can be found in Appendix B. We use the *same* model setting for private training and distance computation. "-" means DP-SGD training.

| AUC | Private Dataset | Public Dataset | Distance |
|---|---|---|---|
| **87.06%** | | HAM10000 | **0.50** |
| 85.53% | HAM10000 | KagSkin | 0.68 |
| 85.40% | | - | - |
| 84.92% | | CIFAR-100 | 0.73 |
| 84.88% | | KagChest | 0.73 |

## 6.3 TRANSFERABILITY: SIMPLE MODELS REMAIN PREDICTIVE

Our empirical evaluation suggests that GSD is transferable over different architectures. In previous experiments, we used the same model architecture for both GSD and the (private) learning algorithm.

We find that the relative GSD ordering of different public datasets is robust across different architectures. For example, **GSD**(ChestX-ray14, KagChest) is consistently smaller than **GSD**(ChestX-ray14, CIFAR-100), no matter what model architecture or parameter-efficient fine-tuning mechanism we choose. Inspired by this finding, we measure GSD with a very simple CNN, which we call a "probe network." It consists of two convolutional layers and one linear layer, with roughly 30,000 parameters. Evaluation results are given in Appendix D.5. They demonstrate that even using a simple CNN, GSD can still derive accurate distance measurement with regard to the utility of public data for private learning tasks. The similarity described by GSD is thus robust against the choice of model architecture.

### 6.4 TASK2VEC MAY GIVE WRONG PREDICTIONS

**Result.** We evaluate the similarity between each public-private dataset pair using Task2Vec Achille et al. (2019). Task2Vec gives similarity results of mixed quality: to highlight one notable failure case, we consider the ChestX-ray14 private dataset in Table 4. The closest dataset is itself. However, following this, CIFAR-100 is as close as KagChest, while it is qualitatively very different from ChestX-ray14 and provides low utility when used as the public dataset. In contrast, GSD orders the quality of these datasets in a manner consistent with their quality. We find similar discrepancies for HAM10000, results are given in the Appendix D.6.

Table 4: Results for GSD vs. Task2Vec. We evaluate ChestX-ray14 using GEP and compute the distance using Task2Vec and GSD. As suggested by Task2Vec, CIFAR-100 should be as close to ChestX-ray14 as KagChest, while it actually provides low utility.

|  | AUC | Task2Vec | GSD |
| --- | --- | --- | --- |
| ChestX-ray14 | 69.02% | 0.052 | 0.15 |
| KagChest | 66.62% | 0.16 | 0.36 |
| - | 64.90% | - | - |
| CIFAR-100 | 48.80% | 0.16 | 0.55 |

## 7 LIMITATIONS AND DISCUSSIONS

We note that, as stated, GSD is not differentially private, as it interacts with the unprotected gradients of the private data. For our scientific question, exploring which public datasets perform well for a particular private task, this is inconsequential. For the algorithmic problem of actually selecting a public dataset for use, one may fear that choosing a public dataset based on GSD may leak sensitive information about the private dataset. However, we remind that the selection of a public dataset is a hyperparameter: in essentially all work on private ML, hyperparameter tuning is performed non-privately, since the privacy impact is considered to be minimal (Papernot & Steinke, 2022; Mohapatra et al., 2022). Thus, for the purpose of our scientific exploration and to be comparable with the majority of the private ML literature, our experiments are performed with non-private computation of GSD. However, we nonetheless discuss differentially private methods for GSD computation in Appendix C. Furthermore, in Appendix D.1, we empirically evaluate the privacy leakage of GSD under a number of membership inference attacks, and even with very strong assumptions on the attacker, they are unable to mount effective attacks. We leave the development of a practical method with rigorous DP guarantees for future work.

## 8 CONCLUSION

A recent line of work explores the power of public data in private machine learning. However, we do not yet have a good understanding of which public datasets are more or less effective for a particular private task, and thus how to select them. We propose a new distance GSD, and empirically demonstrate that lower GSD of a public dataset is strongly predictive of higher downstream utility. Our algorithms require minimal data and are computationally efficient. Additionally, transferability of GSD demonstrates that it is generally model agnostic, allowing one to decouple the public dataset selection and private learning. We further demonstrate that GSD is effective for predicting utility in settings involving both pre-conditioning and second-phase pre-training, and that GSD compares favorably to other measures of dataset distance.

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

## A MISSING PRELIMINARIES

**Definition 3 (Principal Angles (Golub & Loan, 1996)).** *Let $V_1$ and $V_2$ be two orthonormal matrices of $\mathbb{R}^{p \times k}$. The* principal angles $0 \leq \theta_1 \leq \cdots \leq \theta_k \leq \pi/2$ *between two subspaces span($V_1$) and span($V_2$), are defined recursively by*

$$\cos \theta_k = \max_{\mathbf{u_k} \in span(V_1)} \max_{\mathbf{v_k} \in span(V_2)} \mathbf{u'_k v_k}, \ \ subject \ to$$

$$\mathbf{u'_k u_k} = 1, \mathbf{v'_k v_k} = 1, \mathbf{u'_k u_i} = 0, \mathbf{v'_k v_i} = 0, i = 1, ..., k-1$$

That is, the first principal angle $\theta_1$ is the smallest angle between all pairs of unit vectors over two subspaces, the second $\theta_2$ is the second smallest angle, and the rest are similarly defined.

**Definition 4 (($\rho, \eta$)-close).** *A randomized algorithm $\mathcal{A}(.)$ that outputs an approximate distance between to subspaces span($V_1$) and span($V_2$), $\hat{d}(V_1, V_2)$, is an ($\rho, \eta$)-close approximation to the true subspace distance $d(V_1, V_2)$, if they satisfy:*

$$\Pr\left[|\hat{d}(V_1, V_2) - d(V_1, V_2)| \leq \rho\right] \geq 1 - \eta.$$

**Gradient Embedding Perturbation (GEP).** Our theoretical analysis is based on GEP, the state-of-the-art private learning algorithm that leverages public data. Here we briefly introduce their algorithm. GEP involves three steps: 1) it computes a set of the orthonormal basis for the lower-dimensional subspace; 2) GEP projects the private gradients to the subspace derived from step 1, thus dividing the private gradients into two parts: embedding gradients that contain most of the information carried by the gradient, and the remainder are called residual gradients; 3) GEP clips two parts of the gradients separately and perturbs them to achieve differential privacy.

---

**Algorithm 2** Gradient Embedding Perturbation (GEP)

---

**Input:** Private dataset $X_{priv}$, public examples $x_{pub}$, loss function $\mathcal{L}$, model weights $\boldsymbol{\theta}_0$, dimension $k$, learning rate $\eta$, number of iterations $T$, noise multiplier $\sigma_1, \sigma_2$, clip norm $S_1, S_2$
**Output:** Differentially private model $\boldsymbol{\theta}_T$
1: **For** $t = 1$ **to** $T$
2: Compute per-sample gradient matrix $G_t^{priv}$ and public gradient matrix $G_t^{pub}$
3: // Compute an orthonormal basis for the public subspace
4: Initialize $V_k^{pub} \in \mathbb{R}^{k \times p}$ randomly.
5: **For** $i = 1$ **to** $T_{power}$
6: Compute $A = G_t^{pub} V_k^{pub\top}$ and $V_k^{pub} = A^\top V_k^{pub\top}$
7: Orthogonalize $V_k^{pub}$ and normalize row vectors.
8: **EndFor**
9: Delete $G_t^{pub}$ to free memory.
10: // Project the private gradients onto public subspace
11: Compute gradient embedding $W_t = G_t^{priv} V_{k,t}^{pub\top}$ and clip its rows with $S_1$ to obtain $\hat{W}$.
12: Compute residual gradients $R_t = G_t^{priv} - W_t V_{k,t}^{pub}$ and clip its rows with $S_2$ to obtain $\hat{R}$.
13: // Perturb gradient embedding and residual gradient separately
14: Perturb embedding with noise $\boldsymbol{z}_t^{(1)} \sim \mathcal{N}\left(0, \sigma_1^2 \boldsymbol{I}_{k \times k}\right)$: $w_t :=$sum over rows of $\hat{W}_t$, $\hat{w}_t := w_t + \boldsymbol{z}_t^{(1)}$
15: Perturb residual gradient with noise $\boldsymbol{z}_t^{(2)} \sim \mathcal{N}\left(0, \sigma_2^2 \boldsymbol{I}_{k \times k}\right)$: $r_t :=$sum over rows of $\hat{R}_t$, $\hat{r}_t := r_t + \boldsymbol{z}_t^{(2)}$
16: $\hat{v}_t := (\hat{w}_t^\top V_k^{pub} + \hat{r}_t)/n$
17: // Update weights
18: $\boldsymbol{\theta}_{t+1} = \boldsymbol{\theta}_t - \eta \hat{v}_t$
19: **EndFor**

---

## B  EXPERIMENTS SETTING

**Model Architecture.**  As to *pre-conditioning* experiments, for Fashion MNIST, we use a simple convolutional neural network with around 26000 parameters as in Table 5a. For SVHN and CIFAR-10, we use ResNet20 which contains roughly 260,000 parameters. Batch normalization layers are replaced by group normalization layers for different private training, aligning with GEP settings. For ChestX-ray14, we use ResNet152 which has been pretrained on ImageNet1k, a subset of the full ImageNet Deng et al. (2009) dataset. We privately fine-tune its classification layer, which contains around 28,000 parameters. We use the same model architecture for subspace distance computation and GEP private training. As to *second-phase* experiments, we evaluate ChestX-ray14, HAM10000 on ResNet152, DenseNet121, and ViT using various parameter-efficient fine-tuning techniques, we list them in Table 6. We use a simple 2-layer CNN for the probe network, shown in Table 5b.

Table 5: Self-designed model architectures.

(a) Model architecture for Fashion MNIST.

| Layer | Parameters |
|---|---|
| Conv2d | 16 filters of 8x8, stride=2 |
| Maxpooling2d | stride=2 |
| Conv2d | 32 filters 4x4, stride=2 |
| Linear | 32 units |
| Softmax | 10 units |

(b) Model architecture for Probe Network.

| Layer | Parameters |
|---|---|
| Conv2d | 64 filters of 8x8, stride=5 |
| Maxpooling2d | stride=2 |
| Conv2d | 16 filters 4x4, stride=3 |
| Maxpooling2d | stride=2 |
| Linear | 144 units |
| Sigmoid | num_classes |

Table 6: Parameter-efficient fine-tuning mechanisms for different models.

| Model | Fine-tuning mechanism |
|---|---|
| ResNet152 | fc + layer3.32.conv1.weight |
| DenseNet121 | classifier
+ features.denseblock3.denselayer23.conv1.weight
+ features.denseblock3.denselayer24.conv1.weight |
| ViT | LoRA ($\alpha = 8, r = 8$) |

**Dataset Choice.**  CIFAR-10, SVHN and Fashion MNIST are commonly used for evaluation purposes in Computer Vision. For all the evaluations in Table 2, we use the first 2000 images in the test set as public images. For medical datasets, ChestX-ray14 consists of frontal view X-ray images with 14 different classes of lung disease. In our evaluation, there are 78,466 training examples and 20433 testing examples in ChestX-ray14. HAM10000 is composed of 10,000 dermatoscopic images of pigmented lesions. Our choices of public datasets for the four private datasets are described in Table 7. Among them, MNIST-M Ganin et al. (2016) consists of MNIST digits placed on randomly selected backgrounds taken from color photos in the BSDS500 dataset. FLOWER Nilsback & Zisserman (2008) consists of 102 flower categories. KagChest Kermany et al. (2018) imagees were selected from retrospective cohorts of pediatric patients of one to five years old from Guangzhou Women and Children's Medical Center, Guangzhou. It is easy to be obtained from Kaggle so we name it KagChest. SprXRay is another public chest x-ray dataset designed to help predict the age and gender of the patient based on the X-Ray image de Aguiar Kuriki et al. (2023). Kneeos is a dataset Ahmed & Mstafa (2022) that includes X-ray images of knees, which can be used for detecting knee joint issues and grading the severity of knee osteoarthritis using the Kellgren-Lawrence (KL) scale. CheXpert Irvin et al. (2019) is another chest x-ray dataset by Stanford. Similarly for the KagSkin Fanconi (2019), which images of benign skin moles and malignant skin moles. We split the first 300 (for ChestX-ray14, this number is 2000) images in the testset and take them as public. The next 300 (again, for ChestX-ray14, this number is 2000) images are the chose private examples.

**Hyperparameter Setting.**  We use $\epsilon = 2$ and $\delta = 1e - 5$ for all the evaluations. For distance computation, we choose $k = 16$. We follow the hyperparameter setting in the GEP paper for evaluation. In the GEP paper, they didn't evaluate GEP on the ChestX-ray14 dataset. In our

Table 7: Choices of public dataset for private dataset. The four datasets in the first row are private datasets. The datasets listed in the first columns are choices of public datasets. 'X' means we choose the two corresponding datasets as a pair of private/public dataset.

| | CIFAR-10 | SVHN | Fashion MNIST | ChestX-ray14 | HAM10000 |
|---|---|---|---|---|---|
| CIFAR-10 | X | | | | |
| CIFAR-100 | X | X | | X | X |
| SVHN | X | X | | | |
| MNIST-M | | X | | | |
| Fashion MNIST | | | X | | |
| FLOWER | | | X | | |
| MNIST | | | X | | |
| ChestX-ray14 | | | | X | |
| KagChest | | | | X | X |
| HAM10000 | | | | | X |
| KagSkin | | | | | X |
| SprXRay | | | | X | |
| CheXpert | | | | X | |
| Kneeos | | | | X | |

evaluation, we choose $k = 100$ and clip norms are 3 and 1 for original and residual gradients, respectively. The learning rate for the SGD optimizer is set to 0.05. All other hyperparameters are set as default. For second-phase pre-training, we use those public examples to perform supervised learning on trainable parameters. We use Adam optimizer and set $\eta = 3e - 3$ for Transformer-based models and $\eta = 5e - 5$ for CNN-based models. For private fine-tuning, we use SGD optimizer and set $\eta = 0.8$ and clip norm $= 0.1$.

**Compute.** Experiments are conducted on a Linux cloud server with one NVIDIA A100 GPU.

## C PRIVATE DISTANCE MEASUREMENT

While Algorithm 1 has relatively low data exposure (requiring only a single batch of private examples), it is not differentially private. In this section, we give a general algorithm that computes GSD differentially privately: Differentially Private Gradient Subspace Distance (DP-GSD, Algorithm 3). As GSD needs top-$k$ singular vectors from private examples, we derive these singular vectors in a differentially private manner, and the rest of the algorithm remains DP because of post-processing.

---

**Algorithm 3** Differentially Private Gradient Subspace Distance (DP-GSD)

---

**Input:** $m$ private examples $x_{priv}$, $m$ public examples $x_{pub}$, loss function $\mathcal{L}$, model weights $\mathbf{w}_0$, dimension $k$, privacy parameter $\epsilon, \delta$, clip norm $c$
**Output:** Distance between two image datasets $\boldsymbol{d}$

1: // Compute per-sample gradient matrix for private and public
   examples
2: $G_{priv} = \nabla\mathcal{L}(\mathbf{w}_0, x_{priv})$
3: $G_{pub} = \nabla\mathcal{L}(\mathbf{w}_0, x_{pub})$
4: // **Privately** compute top-$k$ subspace of the private gradient
   matrix
5: Clip per-row: $G_{priv} = \mathbf{Clip}(G_{priv}, c)$
6: Compute $V_k^{priv} \leftarrow \mathbf{DPPCA}(G_{priv}, k, \epsilon, \delta)$
7: // Compute top-$k$ subspace of the public gradient matrix
8: $U^{pub}, S^{pub}, V^{pub} \leftarrow \mathbf{SVD}(G_{pub})$
9: // Compute the distance between two subspaces
10: $\boldsymbol{d} = \mathbf{ProjectionMetric}(V_k^{priv}, V_k^{pub})$

---

At a high level, DP-GSD makes one adaptation to GSD: we compute top-$k$ subspace of the private per-sample gradient matrix in a differentially private manner. **DPPCA** in line 6 of Algorithm 3 can be

any Differentially Private Principal Component Analysis (DPPCA), e.g., input perturbation Chaudhuri et al. (2013), subspace perturbation Dwork et al. (2014), exponential mechanism Chaudhuri et al. (2013) and stochastic methods Liu et al. (2022).

We give a theoretical analysis of the privacy and utility guarantee of DP-GSD based on the implementation of DPPCA Chaudhuri et al. (2013), given in Algorithm 4.

---

**Algorithm 4** Differentially Private Principal Component Analysis (DPPCA)

---

    **Input:** $m \times p$ data matrix $X$, dimension $k$, privacy parameter $\epsilon$
    **Output:** $\hat{V}_k$: Top-$k$ subspace of $X$
1: Set $A = \frac{1}{m} X^\top X$
2: Sample $\hat{V}_k = \mathbf{BMF}\left(\frac{m\epsilon}{2} A\right)$

---

To achieve DP, DPPCA randomly samples a $k$-dimensional distribution from the matrix Bingham distribution, which has the following density function:

$$f(V|A, k, p) = \frac{1}{{}_1F_1\left(\frac{1}{2}k, \frac{1}{2}p, A\right)} \exp\left(\mathrm{tr}\left(V^T A V\right)\right) \tag{5}$$

where $V$ is the $p \times k$ subspace and ${}_1F_1\left(\frac{1}{2}k, \frac{1}{2}p, A\right)$ is a normalization factor. We use $\mathbf{BMF}(V)$ in Algorithm 4 to denote this distribution. Thus, we have the following privacy and utility guarantees (proofs in Appendix E):

**Theorem C.1.** *(Privacy Guarantee) Let Algorithm 4 be an implementation of* $\mathbf{DPPCA}$ *in DP-GSD, then DP-GSD is* $\epsilon/c^2$*-differentially priate.*

**Theorem C.2.** *(Utility Guarantee) Let Algorithm 4 be an implementation of* $\mathbf{DPPCA}$ *in DP-GSD, then for $k = 1$, the distance given by DP-GSD, $\hat{d}(V_k^{priv}, V_k^{pub})$ is $(\rho, \eta)$-close to the distance given by GSD, $d(V_k^{priv}, V_k^{pub})$, if we have*

$$m > \frac{pc^2}{\epsilon\alpha(1 - \sqrt{1 - \rho^2})}\left(4\frac{\log(1/\eta)}{p} + 2\log\frac{8\lambda_1}{\rho^2\alpha}\right) \tag{6}$$

*where $\lambda_1$ is the top eigenvalue, $\alpha = \lambda_1 - \lambda_2$ is the eigen-gap, $p$ is the model dimension, $c$ is clip norm and $\epsilon$ is the privacy parameter.*

## D   More Experiments

### D.1   Membership Inference Attack: Cannot Infer Anything

Like other hyperparameter searches, GSD should be used locally and only the best public dataset will be reported (e.g. CIFAR-100 for private CIFAR-10). To claim that this won't reveal sensitive information too much when we use a batch of private data for computing GSD, we evaluate it under membership inference attacks. We choose a well-studied membership inference attack introduced in Nasr et al. (2019) under white box setting and use Adversarial Robustness Toolbox (ART) (Nicolae et al., 2018), a Python library for ML under black box setting.

Based on the recommendations of Nasr et al. (2019), we have the following assumption one the adversary: White-box & Black-box, Stand-alone, Passive and Supervised:

- Black-box refers to the situation where the attacker can only obtain the output. White-box means the attacker can access the full model: intermediate computations, gradients, etc.

- Stand-alone means that GSD is computed in a centralized manner.

- Passive means that the attacker can observe the computation. The attacker cannot be part of the computation in the federated learning setting.

- Supervised: we assume that the attacker knows some of the examples in the private dataset. Note that this is a very strong (even unrealistic) assumption for the attacker. We did this evaluation to illustrate that even though the attacker have such knowledge, the attacker still cannot infer if a new example is in the private data selected for GSD or not.

The white-box membership inference attack, introduced by Nasr et al. (2019), makes use of the information from loss values, gradients, hidden layer activations, etc. They combine these information and supervisely train a binary classifier to tell if a particular image $(x_i, y_i)$ is in $x_{priv}$ or not. Without having access to the model, the black-box attack of Nicolae et al. (2018) trains a binary classifier on the images in a supervised manner.

**Experiment Setting.** Let's say based on the result by GSD: "CIFAR-100 is the best dataset for CIFAR-10" (see Table 2), the adversary wants to figure out if a particular image $(x_i, y_i)$ is used in GSD computation or not (i.e. infer whether $(x_i, y_i)$ in $x_{priv}$. Under white-box setting, we assume that the adversary knows the model we use, i.e. Resnet20. More importantly, we assume that the adversary has already known that $n$ numbers of images we used in $x_{priv}$. Since the adversary knows that CIFAR-10 is the private dataset, we also assume that the adversary has access to all the CIFAR-10 datasets. (still, this is a strong and even unrealistic assumption for the adversary in practice) That is, the adversary has access to all the images in CIFAR-10 and knows $n$ of them are the images we used in $x_{priv}$. The adversary wants to know which of the rest of them are in $x_{priv}$. The test set is composed of the rest of the images in $x_{priv}$ and a same amount of images that are not in $x_{priv}$.

Table 8: Attack success rate of white-box membership inference attack. Note that $|x_{priv}| = 2000$.

|  | n=1000 | n=1800 |
| --- | --- | --- |
| **No GSD values** |  |  |
| Use gradients | 50% | 50% |
| Use hidden layer activations | 50% | 50% |
| **Add GSD values** |  |  |
| Use gradients | 50.25% | 50.31% |
| Use hidden layer activations | 50.05% | 50.06% |

Evaluation results on white-box membership inference attack are given in Table 8. Since GSD computation doesn't require training the model, the model doesn't contain any information about the dataset, i.e., CIFAR-10, even when the adversary knows 1800 images in $x_{priv}$, still cannot infer whether a new image $(x_i, y_i)$ is in $x_{priv}$ or not. Observing these results, we give an even stronger assumption on the adversary: the adversary also knows the public datasets we use, i.e. CIFAR-100 and SVHN (see Table 2), and knows *exactly* which batch we are using. Based on the suggestions of Nasr et al. (2019), we add extra information as follows: the adversary uses those datasets to compute GSD, observes it T times, and adds these GSD values to the binary classifier. But still, the inference is as good as a random guess.

We also evaluate GSD under a black-box membership inference attack, provided by Nicolae et al. (2018). They train a shallow network as a binary classifier in a supervised manner. Results are given in Table 9. Starting the attack with pure images instead of extracting gradients, hidden layer activations seem to have a better attack success rate. But still, this poor attack success rate indicates that GSD can hardly cause privacy leakage.

Table 9: Attack success rate of black-box membership inference attack. Note that $|x_{priv}| = 2000$.

|  | n=1000 | n=1800 |
| --- | --- | --- |
| Attack Success Rate | 51.65% | 52.19% |

### D.2 GRADIENTS ARE IN A LOWER-DIMENSIONAL SUBSPACE.

We evaluate the empirical observation that the stochastic gradients stay in a lower-dimensional subspace during the training procedure of a deep learning model (Gur-Ari et al., 2018; Li et al., 2020), as shown in Figure 3. Results show that only a tiny fraction of singular values are enormous. At the same time, the rest are close to 0, meaning that most of the gradients lie in a lower-dimensional subspace, corresponding to the top singular vectors.

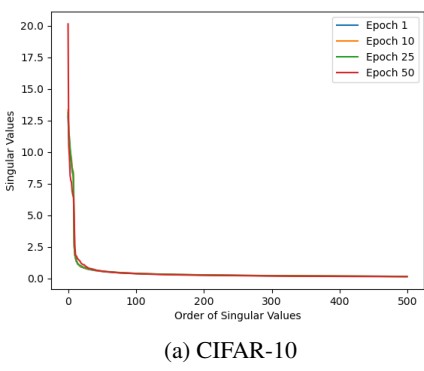
(a) CIFAR-10

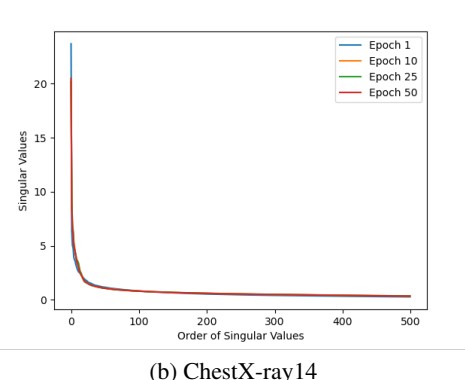
(b) ChestX-ray14

Figure 3: Top 500 singular values in the training procedure using vanilla SGD. Model architectures are in the Appendix B. Only a small fraction of singular values are extremely large while the rest are close to 0, meaning that most of the gradients lie in a lower-dimensional subspace, which corresponds to the top singular vectors.

.

### D.3 MORE PRE-CONDITIONING EVALUATION.

Other than GEP, we also evaluate our method over DP Mirror Descent (Amid et al., 2022), another approach that leverages public data as a preconditioner. We give a short introduction to their algorithm in the Related Work Section 2. Following the evaluation setting in GEP, we have the following experiment results on the ChestX-ray14 dataset (Table 10).

Table 10: DP Mirror Descent evaluation AUC and corresponding distance in descending order. We use the *same* model setting for private training and distance computation. "-" means DP-SGD training without using any public data.

| AUC | Private Dataset | Public Dataset | Distance |
|---|---|---|---|
| **66.83%** | | ChestX-ray14 | **0.15** |
| 65.79% | | SprXRay | 0.32 |
| 65.70% | ChestX-ray14 | CheXpert | 0.32 |
| 64.92% | | KagChest | 0.36 |
| 64.76% | | - | - |
| 50.21% | | CIFAR-100 | 0.55 |

The order of the public datasets is still preserved except for the fact that GEP works slightly better than DP Mirror Descent on larger datasets like ChestX-ray14.

### D.4 MORE SECOND-PHASE PRE-TRAINING EVALUATION.

We evaluate second-phase pre-training and GSD on ChestX-ray14 and HAM10000 using ResNet152, DenseNet121, and ViT. Aside from the results we presented, we show the rest of the results here.

Table 11: Second-Phase evaluation results and corresponding distance on ChestX-ray14 in descending order. Detailed settings can be found in Appendix B. We use the *same* model setting for private training and distance computation. "-" means vanilla DP-SGD training.

(a) ResNet152

| AUC | Private Dataset | Public Dataset | Distance |
|---|---|---|---|
| **67.48%** | | ChestX-ray14 | **0.31** |
| 67.27% | ChestX-ray14 | KagChest | 0.34 |
| 66.82% | | - | - |
| 66.57% | | CIFAR-100 | 0.39 |

(b) DenseNet121

| AUC | Private Dataset | Public Dataset | Distance |
|---|---|---|---|
| **67.53%** | | ChestX-ray14 | **0.33** |
| 67.47% | ChestX-ray14 | - | - |
| 67.40% | | KagChest | 0.37 |
| 67.28% | | CIFAR-100 | 0.40 |

Table 12: Second-Phase evaluation results and corresponding distance in descending order. We use ViT and LoRA fine-tuning. We use the *same* model setting for private training and distance computation. Detailed settings can be found in Appendix B. "-" means DP-SGD training.

| AUC | Private Dataset | Public Dataset | Distance |
|---|---|---|---|
| **72.99%** | | ChestX-ray14 | **0.44** |
| 71.86% | ChestX-ray14 | KagChest | 0.59 |
| 70.93% | | - | - |
| 70.84% | | CIFAR-100 | 0.98 |

Table 13: Second-Phase evaluation results and corresponding distance on HAM10000 in descending order. Detailed settings can be found in Appendix B. We use the *same* model setting for private training and distance computation. "-" means vanilla DP-SGD training.

(a) ResNet152

| AUC | Private Dataset | Public Dataset | Distance |
|---|---|---|---|
| **86.83%** | | HAM10000 | **0.48** |
| 85.95% | HAM10000 | KagSkin | 0.65 |
| 85.55% | | - | - |
| 85.49% | | CIFAR-100 | 0.70 |
| 85.41% | | KagChest | 0.70 |

(b) ViT

| AUC | Private Dataset | Public Dataset | Distance |
|---|---|---|---|
| **84.94%** | | HAM10000 | **0.50** |
| 81.23% | | KagSkin | 0.76 |
| 78.65% | HAM10000 | KagChest | 0.93 |
| 77.07% | | CIFAR-100 | 0.97 |
| 73.67% | | - | - |

## D.5 TRANSFERABILITY: SIMPLE MODELS REMAIN PREDICTIVE

GSD is transferable across different architectures and the relative GSD ordering of public datasets remains consistent. We present the results in Table 14.

Table 14: Transferability evaluation results. The left-most column denotes each pair of private-public datasets, e.g. (Xray, Xray) means we take ChestX-ray14 as a private dataset, split part of its testset and take those images as public. Detailed settings can be found in Appendix B. The first row denotes different model architectures. "Probe" is a simple CNN with around 30,000 parameters. ResNet152* and ResNet152** use different parameter-efficient fine-tuning settings. We use the same model for each Distance-Accuracy. The results show that this distance given by GSD is generally robust across different algorithms (pre-conditioning or second-phase pre-training) and different model architectures (from simple Probe to ViT). A smaller distance indicates that this public dataset is more similar to the private one, thus leveraging this public dataset for private learning will result in better accuracy.

|  | Probe | ResNet152* | ResNet152** | DenseNet121 | ViT |
|---|---|---|---|---|---|
| Task |  | Pre-conditioning | Second-phase | Second-phase | Second-phase |
|  |  | Distance \| Accuracy | | | |
| (Xray, Xray) | **0.39** | **0.15 \| 69.02%** | **0.31 \| 67.48%** | **0.33 \| 67.53%** | **0.44 \| 72.99%** |
| (Xray, Chest) | 0.52 | 0.36 \| 66.62% | 0.34 \| 67.27% | 0.37 \| 67.40% | 0.59 \| 71.86% |
| (Xray, CIFAR) | 0.58 | 0.55 \| 48.80% | 0.39 \| 66.57% | 0.40 \| 67.28% | 0.98 \| 70.84% |
| (HAM, HAM) | **0.42** | - | **0.48 \| 86.83%** | **0.50 \| 87.06%** | **0.50 \| 84.94%** |
| (HAM, Skin) | 0.55 | - | 0.65 \| 85.95% | 0.68 \| 85.53% | 0.76 \| 81.23% |
| (HAM, CIFAR) | 0.67 | - | 0.70 \| 85.49% | 0.73 \| 84.92% | 0.97 \| 77.07% |
| (HAM, Chest) | 0.76 | - | 0.70 \| 85.41% | 0.73 \| 84.88% | 0.93 \| 78.65% |

### D.6 MORE VS. TASK2VEC.

We evaluate the similarity between each public-private dataset pair using Task2Vec. The results on HAM10000 public dataset are presented here using cluster map in Figure 4. The closest datasets are itself, and the HAM10000 public dataset. However, following this, the closest dataset is KagChest, which is qualitatively very different from HAM10000 (chest x-rays versus skin lesions) and provides low utility when used as the public dataset (see Table 14). In particular, KagSkin (another skin disease dataset) is qualitatively closer to HAM10000 and provides higher utility when used as a public dataset, yet Task2Vec assigns it a greater distance than KagChest. In contrast, GSD orders the quality of these datasets in a manner consistent with their quality.

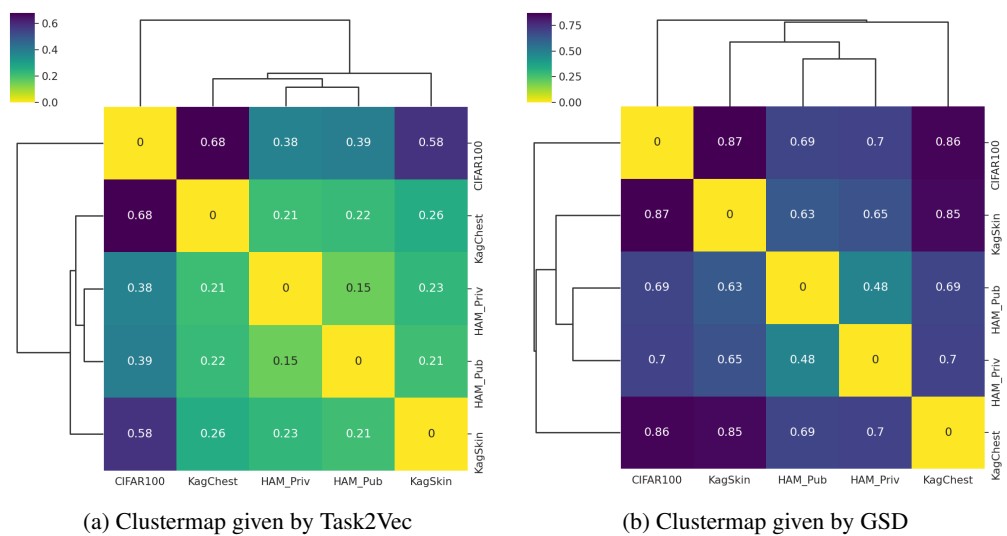

(a) Clustermap given by Task2Vec    (b) Clustermap given by GSD

Figure 4: Clustermaps given by Task2Vec (left) and GSD(right) for HAM10000 and corresponding public datasets. The lines on the top and the left denote the similarity of a pair of datasets. The numbers in the grid are the similarity distance for a pair of datasets. Although Task2Vec gives the correct prediction for (HAM$_{public}$, HAM$_{private}$) pair, it ***incorrectly*** predicts the second-close similarity, where it believes KagChest is the second-close dataset for HAM10000, while they should be totally irrelevant.

### D.7 COMBINATIONS OF DATASETS BETTER THAN ONE: GENERALLY BUT NOT NECESSARILY TRUE

While some studies point out that pre-training on multiple diverse datasets can lead to better performance on downstream NLP tasks compared to pre-training on a single dataset (Brown et al., 2020), our experiment results show that under small amount of accessible public data assumption, this conclusion is generally but not necessarily true, as shown in Table 15.

Table 15: GEP evaluation AUC and corresponding distance in descending order. We use the *same* model setting for private training and distance computation. "-" means DP-SGD training without using any public data.

| AUC | Private Dataset | Public Dataset | Distance |
|---|---|---|---|
| **69.02%** | | ChestX-ray14 | **0.15** |
| 67.83% | | SprXRay + CheXpert | 0.30 |
| 67.31% | | SprXRay | 0.32 |
| 67.29% | | CheXpert | 0.32 |
| 67.12% | ChestX-ray14 | CheXpert + CIFAR-100 | 0.33 |
| 66.62% | | KagChest | 0.36 |
| 65.26% | | Kneeos | 0.38 |
| 64.90% | | - | - |
| 48.80% | | CIFAR-100 | 0.55 |

Using SprXRay or CheXpert as public datasets for private learning can result in higher accuracy compared to the DP-SGD baseline. Combining these datasets can further improve accuracy. However, if a "bad" public dataset like CIFAR-100 is combined with ChestX-ray14, it can negatively impact the overall performance.

## E MISSING PROOFS

In this section, we present proofs for Section 4.1.

**Lemma 4.1 (Reconstruction Error).** For GEP, let $\mathbf{G}_{priv}$, $V_k^{pub}$, $V_k^{priv}$ be the gradient matrix and top-$k$ gradient subspace from public or private examples at step t, respectively. Then we have the spectral norm of reconstruction error

$$\|\mathbf{R}\|_2 \leq \sqrt{2}s_{1,t}\mathbf{GSD}(V_k^{priv}, V_k^{pub}) + s_{k+1,t} \tag{7}$$

where $\mathbf{R} = \mathbf{G}_{priv} - \mathbf{G}_{priv}V_k^{pub}V_k^{pub\top}$ is the reconstruction error of private gradient matrix $\mathbf{G}_{priv}$ using public examples, $s_{1,t} \geq ... \geq s_{k,t} \geq ...$ are the singular values of $\mathbf{G}_{priv}$, $\mathbf{GSD}(V_k^{priv}, V_k^{pub})$ is the gradient subspace distance given by our algorithm.

*Proof.* We have

$$\mathbf{R} = \mathbf{G}_{priv} - \mathbf{G}_{priv}\Pi_k^{pub}$$
$$= \mathbf{G}_{priv} - \mathbf{G}_{priv}\Pi_k^{priv} + \mathbf{G}_{priv}\Pi_k^{priv} - \mathbf{G}_{priv}\Pi_k^{pub} \tag{8}$$

$$\Rightarrow \quad \|\mathbf{R}\|_2 - \left\|\mathbf{G}_{priv}(\Pi_k^{pub} - \Pi_k^{priv})\right\|_2 \leq \left\|\mathbf{G}_{priv}\left(\mathbb{I} - \Pi_k^{priv}\right)\right\|_2 \tag{9}$$

$$\Rightarrow \quad \|\mathbf{R}\|_2 \leq \underbrace{\left\|\mathbf{G}_{priv}(\Pi_k^{pub} - \Pi_k^{priv})\right\|_2}_{D_1} + \underbrace{\left\|\mathbf{G}_{priv}\left(\mathbb{I} - \Pi_k^{priv}\right)\right\|_2}_{D_2} \tag{10}$$

where $\Pi_k^{pub} = V_k^{pub}V_k^{pub\top}$ denotes the orthogal projection to the subspace of $\text{span}(V_k^{pub})$, $\Pi_k^{priv} = V_k^{priv}V_k^{priv\top}$ denotes the orthogal projection to the subspace of $\text{span}(V_k^{priv})$.

For $D_2$, recall that the Eckart–Young–Mirsky theorem (Eckart & Young, 1936) shows that the best rank-$k$ approximation of $\mathbf{G}_{priv}$ is given by its top-$k$ reconstruction using SVD. Therefore, we have

$$D_2 = \left\| \mathbf{G}_{priv} \left( \mathbb{I} - \Pi_k^{priv} \right) \right\|_2$$

$$= \left\| \sum_{i=1}^{p} s_i u_i v_i^\top - \sum_{i=1}^{k} s_i u_i v_i^\top \right\|_2 \tag{11}$$

$$= \left\| \sum_{i=k+1}^{p} s_i u_i v_i^\top \right\|_2$$

$$= s_{k+1}$$

For $D_1$, the definition of projection metric (Definition 3) shows that

$$\mathbf{GSD}^2(V_k^{priv}, V_k^{pub}) = k - (\cos^2 \theta_1 + ... + \cos^2 \theta_k)$$

$$\overset{(a)}{=} k - \mathrm{Tr} \left( V_k^{priv\top} V_k^{priv} V_k^{pub\top} V_k^{pub} \right) \tag{12}$$

$$\overset{(b)}{=} \frac{1}{2} \left\| \Pi_k^{pub} - \Pi_k^{priv} \right\|_F^2$$

(a) and (b) hold according to Equation 5.4 of Conway et al. (2002).

Therefore, we have

$$D_1 = \left\| \mathbf{G}_{priv}(\Pi_k^{pub} - \Pi_k^{priv}) \right\|_2$$

$$\leq \| \mathbf{G}_{priv} \|_2 \left\| \Pi_k^{pub} - \Pi_k^{priv} \right\|_2 \tag{13}$$

$$\leq s_1 \left\| \Pi_k^{pub} - \Pi_k^{priv} \right\|_F$$

$$= \sqrt{2} s_1 \mathbf{GSD}(V_k^{priv}, V_k^{pub})$$

Combining $D_1$ and $D_2$, we have

$$\| \mathbf{R} \|_2 \leq \left\| \mathbf{G}_{priv}(\Pi_k^{pub} - \Pi_k^{priv}) \right\|_2 + \left\| \mathbf{G}_{priv} \left( \mathbb{I} - \Pi_k^{priv} \right) \right\|_2 \tag{14}$$

$$\leq \sqrt{2} s_{1,t} \mathbf{GSD}(V_k^{priv}, V_k^{pub}) + s_{k+1,t}$$

Thus we know that GSD bounds the reconstruction error at step $t$. $\square$

**Theorem 4.2 (Excess Risk).** Assume that the loss $L(\mathbf{w}) = \frac{1}{n} \sum_{i=1}^{n} \ell(\mathbf{w}, z_i)$ is 1-Lipschitz, convex, and $\beta$-smooth. Let $\mathbf{w}^* = \mathrm{argmin}_{w \in \mathcal{W}} L(\mathbf{w})$. The excess risk of GEP obeys

$$\mathbb{E}[L(\overline{\mathbf{w}})] - L(\mathbf{w}^*) \leq O\left( \frac{\sqrt{k \log(1/\delta)}}{n\epsilon} \right) + O\left( \frac{\sqrt{p \log(1/\delta)}}{n\epsilon} \overline{\mathbf{d}} \right) \tag{15}$$

where GEP is $(\epsilon, \delta)$-DP (see Appendix A). Here we set $\eta = \frac{1}{\beta}, T = \frac{n\beta\epsilon}{\sqrt{p}}, \overline{\mathbf{w}} = \frac{1}{T} \sum_{t=1}^{T} \mathbf{w}_t$, $\overline{\mathbf{d}} = \frac{1}{T} \sum_{t=1}^{T} d_t^2, d_t = \sqrt{2} s_{1,t} \mathbf{GSD} + s_{k+1,t}$, and $\mathbf{GSD}$, $s$ are the gradient subspace distance and singular values of the gradient matrix at step t.

*Proof.* Theorem 3.3 of Yu et al. (2021) (see Appendix A) shows that the excess risk of GEP obeys

$$\mathbb{E}[L(\overline{\mathbf{w}})] - L(\mathbf{w}^*) \leq O\left( \frac{\sqrt{k \log(1/\delta)}}{n\epsilon} \right) + O\left( \frac{\overline{r} \sqrt{p \log(1/\delta)}}{n\epsilon} \right) \tag{16}$$

where $\overline{r} = \frac{1}{T} \sum_{t=0}^{T-1} r_t^2$ and $r_t = \| \mathbf{G}_{priv} - \mathbf{G}_{priv} V_k^{pub} V_k^{pub\top} \|_2$ is the reconstruction error at step $t$.

From Lemma 4.1 we know that $r_t \leq d_t$ at each step $t$, thus completing the proof. $\square$

**Lemma E.1.** *(Theorem 6 of Chaudhuri et al. (2013)) DPPCA (Algorthim 4) is $\epsilon$-differentially private.*

**Theorem C.1 (Privacy Guarentee).** Let Algorithm 4 be an implementation of **DPPCA** in DP-GSD, then DP-GSD is $\epsilon/c^2$-differentially priate.

*Proof.* Let $x_{priv}$ be $m$ private examples. $G_{priv} = \mathbf{Clip}(\nabla\mathcal{L}(\mathbf{w}_0, x_{priv}), c) \in \mathcal{R}^{m \times p}$ is per-sample gradient matrix and $A = \frac{1}{m}G_{priv}G_{priv}^\top$. We sample top-$k$ eigenvectors from the matrix Bingham distribution (Chikuse, 2003):

$$f(V|A, k, p) = \frac{1}{{}_1F_1\left(\frac{1}{2}k, \frac{1}{2}p, A\right)} \exp\left(\operatorname{tr}\left(V^T A V\right)\right) \tag{17}$$

with $A = \frac{m\epsilon}{2c^2}A$. We show that this is the exponential mechanism applied to the score function $q(V; x) = mv^T A v$.

Consider the neighboring data $X'_{priv} = [x_1, ..., x'_i, ..., x_m]$ that differ from $X_{priv}$ with one data example $x'_i$. Let $G'_{priv} = \mathbf{Clip}(\nabla\mathcal{L}(\mathbf{w}_0, x'_{priv}), c)$ and $A' = \frac{1}{m}G'^\top_{priv}G'_{priv}$. We have

$$\begin{aligned}
\Delta q = \max \left|mv^T A v - mv^T A' v\right| &\leq \left|v^T(g_i g_i^T - g'_i g_i'^\top)v\right| \\
&\leq \left|\|v^T g_i\|\|^2 - \|v^T g'_i\|^2\right| \\
&\leq \left|\|v^T \nabla\mathcal{L}(\mathbf{w}_0, x_i)\|\|^2 - \|v^T \nabla\mathcal{L}(\mathbf{w}_0, x'_i)\|^2\right| \\
&\leq c^2
\end{aligned} \tag{18}$$

Therefore, from E.1, we know that **DPPCA** in DP-GSD (Algorithm 3) is $\epsilon/c^2$-differentially private. Thus, DP-GSD is $\epsilon/c^2$-differentially private because of post-processing. $\qquad\square$

**Lemma E.2.** *(Theorem 7 of Chaudhuri et al. (2013)) Let $k = 1$, the private gradient subspace $V_k^{priv}$ in GSD and the private gradient subspace $\hat{V}_k^{priv}$ from Algorithm 4 satisfy*

$$\Pr\left(\left|\left\langle V_k^{priv}, \hat{V}_k^{priv}\right\rangle\right| > \rho\right) \geq 1 - \eta \tag{19}$$

*if we have*

$$m > \frac{p}{\epsilon\alpha(1-\rho)}\left(4\frac{\log(1/\eta)}{p} + 2\log\frac{8\lambda_1}{(1-\rho^2)\alpha}\right) \tag{20}$$

*where $\lambda_1$ is the top eigenvalue, $\alpha = \lambda_1 - \lambda_2$ is the eigen-gap, $p$ is the model dimension and $\epsilon$ is the privacy parameter.*

**Theorem C.2 (Utility Guarantee)** Let Algorithm 4 be an implementation of **DPPCA** in DP-GSD, then for $k = 1$, the distance given by DP-GSD, $\hat{d}(V_k^{priv}, V_k^{pub})$ is $(\rho, \eta)$-close to the distance given by GSD, $d(V_k^{priv}, V_k^{pub})$, if we have

$$m > \frac{pc^2}{\epsilon\alpha(1-\sqrt{1-\rho^2})}\left(4\frac{\log(1/\eta)}{p} + 2\log\frac{8\lambda_1}{\rho^2\alpha}\right) \tag{21}$$

where $\lambda_1$ is the top eigenvalue, $\alpha = \lambda_1 - \lambda_2$ is the eigen-gap, $p$ is the model dimension, $c$ is clip norm and $\epsilon$ is the privacy parameter.

*Proof.* Let $V_k^{priv}$ be the private gradient subspace in GSD and $\hat{V}_k^{priv}$ be the private gradient subspace in DP-GSD, $d(V_k^{priv}, \hat{V}_k^{priv})$ be the projection metric distance between two subspaces. From Lemma E.2, we have

$$\begin{aligned}
\Pr\left(\left|\left\langle V_k^{priv}, \hat{V}_k^{priv}\right\rangle\right| > \mu\right) &= \Pr\left(d(V_k^{priv}, \hat{V}_k^{priv}) \leq \sqrt{1-\rho^2}\right) \geq 1 - \eta \\
&\xrightarrow{(a)} \Pr\left(\left|\hat{d}(V_k^{priv}, V_k^{pub}) - d(V_k^{priv}, V_k^{pub})\right| \leq \sqrt{1-\mu^2}\right) \geq 1 - \eta
\end{aligned} \tag{22}$$

(a) holds because of the triangle inequality of projection metric (Ham & Lee, 2008). Substituting $\sqrt{1-\mu^2}$ with $\rho$, we have $\hat{d}(V_k^{priv}, V_k^{pub})$ is $(\rho, \eta)$-close to $d(V_k^{priv}, V_k^{pub})$. $\qquad\square$

