# OpenReview forum: "Choosing Public Datasets for Private Machine Learning via Gradient Subspace Distance"
_ICLR.cc/2024/Conference — Submitted to ICLR 2024_

### Official Review · Reviewer_VtTm · 2023-10-24

**Soundness:** 2 fair
**Presentation:** 3 good
**Contribution:** 1 poor
**Rating:** 3
**Confidence:** 3

**Summary:**

Considering training a network privately via differential privacy, this work adopts a two-phase pre-training followed by a low-dimensional adaptation as the training pipeline to achieve better performance. Specially, both the second phase pre-training and low-dimensional adaptation are executed suing a public dataset selected from plenty of candidates via the proposed gradient subspace distance measure. The authors have conducted experiments on various architectures and datasets demonstrate the effectiveness of the proposed method.

**Strengths:**

i) A high-quality basis would be crucial for the performance of low-dimensionally projected DP-SGD. This work manages to achieve this goal by identifying the most suitable public dataset from a group of candidate for computing the basis.

ii) The paper provides analysis to justify why choosing a basis with a smaller gradient subspace distance is beneficial in the context of low-dimensionally projected optimization.

**Weaknesses:**

i) Computing the basis using private dataset compromises privacy, as the basis is directly depending on the private data. Although the authors argue that this can be considered as tuning hyperparameters, such an argument is unconvincing. In particular, this work does not conserve privacy for basic hyperparameter-tuning such as learning rate. Compared to related works or baselines, the proposed method definitely loses more privacy .

ii) The authors claim to focus on smaller public datasets due to computational resource limitations. For instance, instead of pre-training a network in CIFAR100, a batch of CIFAR100 data is selected for the projected DP-SGD. However, this argument seems weak. In my experience, projected DP-SGD is resource -intensive due to the basis calculation and gradient matrix storage. I suspect that conducting PEFT on the entire CIFAR100 dataset followed by vanilla PEFT DP-SGD could be faster and yield better results. This raises questions about the motivation and necessity of devising such a multi-stage training pipeline. The authors could conduct some additional experiments and provide details on running time, memory usage and configurations to justify the merits of their framework.

iii) The best utility gains reported in the experiments are primarily achieved when the private dataset is used as the public dataset. This result is trivial and does not demonstrate the necessity of employing GSD. Additionally, only a limited number of candidates are reported in each experiment, making it is unclear whether the GSD order is aligned well with the actual utility gain.

iv) Most of the utility gains are marginal. I also note the reported results are poor given that the networks are pre-trained. Specifically, the best results of CIFAR10 and FMNIST are worse than or only comparable to some basic baselines, e.g. [1].  Although the original paper of this baseline does not report the results of $\epsilon=2$, the authors can run its code to verify this.

[1] Tramer & Boneh, DIFFERENTIALLY PRIVATE LEARNING NEEDS BETTER FEATURES (OR MUCH MORE DATA), 2021.

**Questions:**

In addition to my questions in the weaknesses section, I have the following questions:

i) Is it consistently the case that CIFAR100 is a better public dataset for CIFAR10 than CIFAR10 itself, regardless of variations in batch size and the data points selected for basis calculation?

ii) Based on the experimental results, it seems that visually similar datasets usually make the best choices. Could Frechlet inception distance, which is widely used in generative models, serve as a replacement for GSD?

---

> ### Author Response · Authors · 2023-11-15
> **Thank you for your constructive feedback**
>
> We thank the reviewer for the constructive feedback. We are encouraged that the reviewer also agrees on the importance of such a scientific question: identifying a high-quality basis for private machine learning, and also agrees that our work manages to give a useful solution. Here, we explain in detail how we address your comments.

---

> > ### Author Response · Authors · 2023-11-15
> > **Address weakness iii**
> >
> > We disagree that the finding is "trivial": indeed, if it were trivial, then it would always be true that using public data drawn from the same distribution as the private data would be ideal. However, in one of our more surprising results, we find this is _not_ true -- that public data from a separate distribution may be better than the original one! For example, CIFAR-100 is consistently a better dataset for CIFAR-10 than itself, and this answers the reviewer’s question i). Furthermore, public data from the same distribution as the private dataset may not always be available: consequently, GSD is useful in determining which alternative dataset one should use. See another example: at first we also assume that MNIST-M (colorful MNIST), is a good public dataset for SVHN, as they are all images of digits. BUT GSD shows that MNIST-M is even worse than CIFAR-100!

---

> > ### Author Response · Authors · 2023-11-15
> > **Address weakness iv**
> >
> > First, the results on CIFAR10 and FMNIST are achieved without pre-training---we DP train a Resnet20 from _scratch_. Second, we respectfully disagree with the argument that the utility gains are insignificant. For instance, when examining the results of ChestX-ray14 dataset, our approach demonstrates a significant ~3% accuracy improvement over the DP-SGD baseline. This improvement is particularly crucial in clinical environments. Furthermore, this paper's primary contribution lies in identifying and addressing an important scientific question: determining which public datasets perform well for specific private tasks. Our solution to this problem is both straightforward and highly effective.

---

> > ### Author Response · Authors · 2023-11-15
> > **Answer question ii**
> >
> > We appreciate the reviewer for highlighting this interesting metric. Visually similar datasets may not always be the most suitable choice for private ML. For instance, while chest x-ray datasets like CheXpert provide utility gains, the Kneeos dataset, consisting of knee X-rays, also offers utility. Conversely, the MNIST-M dataset, which is similar to SVHN as they both consist of images of digits, provides the lowest utility. Task2Vec, a method known for measuring dataset similarity in nature, is unable to accurately rank datasets utility (refer to Section 6.4). Second, it is unclear if other metrics can also enjoy the same properties, i.e., ordering of GSD is preserved over training (Section 4.2), transferability (Section 6.3). That said, that is exactly the main contribution of this paper: we propose this metric, and validate that it is suitable for selecting the best public dataset via thorough empirical validations and some theoretical analysis.

---

> ### Author Response · Authors · 2023-11-15
> **Address weakness i**
>
> We agree that, _in theory,_ the computation of GSD does cause privacy leakage.  However, our empirical evaluation using membership inference attacks reveals that an overestimated adversary has a success rate similar to _random guessing_ in practice. Based on realistic assumptions, the additional information an adversary can obtain from GSD is limited to "CIFAR-100 is the chosen public dataset for CIFAR-10." This information is difficult to construct a meaningful attack. Therefore, we further assume a powerful yet unrealistic adversary: we assume that the adversary knows: 1) the model used by GSD and all the related computations such as gradients, hidden layer activation, 2)  90% of the private data examples used by GSD, and all the rest private examples in the private dataset. The adversary tries to use membership inference attacks to tell if a particular image $x_i$ is used by GSD (i.e., see if $x_i$ is in the rest 10%). We apply one of the most well-studied and well-cited white-box membership inference attacks and the experiments show that: even under this overly estimated and powerful adversary, the attack success rate is as good as random guesses (see Appendix D.1 for more details).
>
> Additionally, we nevertheless discuss DP methods for GSD computation in Appendix C. We hope that this work will serve as a starting point and inspire future research to effectively solve this scientific question with differential privacy.

---

> ### Author Response · Authors · 2023-11-15
> **Address weakness ii**
>
> The reviewer has raised concerns about the claims we made in the paper regarding computational resource limitations. We address these concerns with the following points:
>
> * Indeed we focus on the scenario where practitioners only have a small amount of public datasets. Due to 1) this is consistent with previous literature setting [1][2] 2) this setting is applicable in certain key settings. For example, consider a clinical environment. Due to privacy and safety concerns, it may be inappropriate to use large, uncurated Internet datasets as public data (e.g., such a dataset may be infected by a data poisoning attack). Instead, the practitioner may have a small set of trusted public datasets. In our experiments, we set the number of public data examples to 2000 or 300, depending on the size of the private dataset.
> * Under this smaller public datasets setting, projected DP-SGD is not resource-intensive. It is indeed less time-efficient than “public pre-training, DP fine-tuning”, but all the experiments in this paper can be done on a single NVIDIA A100 GPU with 40GB VRAM. More importantly, as indicated in previous studies on public-data-assisted private ML, projected DP-SGD does improve accuracy. This is why this work is based on them and efficiently addresses the problem of selecting the best public dataset.
> * When discussing computation resource limitations, we refer to large-scale pre-training on datasets like ImageNet, JFT-300, or even foundation models. These computational resource requirements are not on the same scale as the projected DP-SGD. Most practitioners can neither afford the computation resources for such pre-training, nor afford the time, effort and money to collect and create such a dataset. That’s why most practitioners will turn to off-the-shelf pre-trained models, and this advocates the necessity of devising such a “multi-stage training pipeline”. In addition, irrelevant pre-training datasets do not improve accuracy [3][4], highlighting the need for GSD to identify the best public datasets .
> * Additionally, if the reviewer agrees with the setting together with the reasoning we give, “PEFT on the entire CIFAR100 dataset followed by vanilla PEFT DP-SGD” would be inappropriate because we cannot assume that we have enough public examples available.
> * To make this more clear, the “multi-stage training pipeline” is designed as an alternative to “PEFT, vanilla PEFT DP-SGD”, based on the reasoning we give aforementioned. Projected DP-SGD is one of the pre-conditioning methods. We also evaluated other pre-conditioning methods other than projected DP-SGD (see Appendix D.3) and GSD also works. That said, GSD works for all kinds of public-data-assist private ML methods, based on our experiments.
>
> [1] Da Yu, Huishuai Zhang, Wei Chen, and Tie-Yan Liu. Do not let privacy overbill utility: Gradient embedding perturbation for private learning. In 9th International Conference on Learning Representations, ICLR 2021, Virtual Event, Austria, May 3-7, 2021. OpenReview.net, 2021. URL https://openreview.net/forum?id=7aogOj_VYO0.
>
> [2] Yingxue Zhou, Steven Wu, and Arindam Banerjee. Bypassing the ambient dimension: Private sgd with gradient subspace identification. In International Conference on Learning Representations, 2021. URL https://openreview.net/forum?id=7dpmlkBuJFC.
>
> [3] Radford, A., Kim, J. W., Hallacy, C., Ramesh, A., Goh, G., Agarwal, S., Sastry, G., Askell, A., Mishkin, P., Clark, J., Krueger, G., & Sutskever, I. (2021). Learning transferable visual models from natural language supervision. In arXiv [cs.CV]. http://arxiv.org/abs/2103.00020
>
> [4] Da Yu, Sivakanth Gopi, Janardhan Kulkarni, Zinan Lin, Saurabh Naik, Tomasz Lukasz Religa, Jian Yin, and Huishuai Zhang. Selective pre-training for private fine-tuning, 2023.

---

> ### Comment · Reviewer_VtTm · 2023-11-20
> **Official Comment by Reviewer**
>
> Thank the authors for answering my questions and providing further information.
>
> > Response to weakness i
>
> This response doesn't alleviate my concern. Many (membership inference attacks) MIAs have been evaluated under scenarios of limited data and/or overfitted networks. Their performance on realistically trained networks is not perfect. Theoretically, this work proposes a method that releases more privacy. I'm not convinced that the failure of MIAs to prove non-hazardous privacy leakage is sufficient, since MIAs are not robust themselves. Additionally, the choice of a public dataset can sometimes directly release key information, such as when selecting a "tumor dataset" to assist in the private training.
>
> > Response to weakness ii
>
> The author should clarify that this work focuses on the cases where large scale public datasets are not available, and remove the previous claim that the focus is on smaller public datasets due to computational resource limitations.
>
> > Response to weakness iii
>
> It is interesting to note that CIFAR100 is more effective than CIFAR10 for computing the gradient subspace of CIFAR10, but there is only a single example. In contrast, in the most settings, the private datasets are their own best public counterparts. Nevertheless, I agree that a metric predicting the utility gain of incorporating a public dataset is useful. But I believe the current experiments are very limited and do not adequately evaluate the robustness of the proposed metric.
>
> > Response to weakness iv
>
> The reference I provided describes a network with a handcrafted feature extractor and does not use any public data. A comparison would prove whether it is worthwhile to sacrifice some privacy (due to public data selecting) in exchange for significantly better utility.
>
> Since my concerns largely remain, I would like to maintain the score by far.

---

> > ### Author Response · Authors · 2023-11-21
> >
> > Thank the reviewer for the feedback.
> >
> > > Weakness i
> >
> > We agree with the reviewer's suggestion that a theoretical guarantee is needed for privacy, as our current guarantee is based only on heuristics.  However, as discussed in our paper, for our scientific question, exploring which public datasets perform well for a particular private task, this is inconsequential.  In our paper, we discuss some follow-up work that addresses how to select private datasets in a privacy-preserving manner. making it unclear if it can be considered as "knowing the utility in prior" due to the significant time cost.  That said, we hope our work initializes further exploration into this scientific question and provides insights for future work that develops a practical method with rigorous DP guarantee.
> >
> > > Weakness ii
> >
> > We appreciate the reviewer for pointing out this issue. We will revise and clarify this claim accordingly.
> >
> > > Weakness iii
> >
> > We appreciate the reviewer's acknowledgment of the importance of our work's purpose. It makes sense that for most of the private datasets, the best public dataset is itself. However, our method also works when we only have accessible _out-of-distribution_ data. We empirically validate this across various pre-conditioning/pre-training methods, datasets, model architectures, and number of available public examples. We also give theoretical analysis to explain why GSD can be a good indicator of public datasets utility. Furthermore, we demonstrate that GSD's transferability showcases its robustness. That said, we believe the current results presented in this paper are sufficient to validate the usefulness of GSD. Nevertheless, we are happy to discuss with the reviewer what kind of additional evaluation is needed to further validate the robustness of GSD.
> >
> > > Weakness iv
> >
> > Considering it is now near the end of 2023, we believe, to the best of our knowledge, most works studying improving the utility of differentially private machine learning, turn to public data and end-to-end deep models[1][2][3]. Even the first author of the mentioned reference has shifted their research towards studying private machine learning with public data. Additionally, handcrafted feature extractors have limitations as they cannot generalize well to complex tasks. This is why researchers are turning to deep models and public data. Notably, the paper mentioned by the reviewer, includes a section titled "TRANSFER LEARNING: BETTER FEATURES FROM PUBLIC DATA".
> >
> > [1] Da Yu, Huishuai Zhang, Wei Chen, and Tie-Yan Liu. Do not let privacy overbill utility: Gradient embedding perturbation for private learning. In 9th International Conference on Learning Representations, ICLR 2021, Virtual Event, Austria, May 3-7, 2021. OpenReview.net, 2021. URL https://openreview.net/forum?id=7aogOj_VYO0.
> >
> > [2] Yingxue Zhou, Steven Wu, and Arindam Banerjee. Bypassing the ambient dimension: Private sgd with gradient subspace identification. In International Conference on Learning Representations, 2021. URL https://openreview.net/forum?id=7dpmlkBuJFC.
> >
> > [3] Ganesh, Arun, Mahdi Haghifam, Milad Nasr, Sewoong Oh, Thomas Steinke, Om Thakkar, Abhradeep Thakurta, and Lun Wang. "Why Is Public Pretraining Necessary for Private Model Training?." ICML 2023.

---

### Official Review · Reviewer_heJ7 · 2023-11-01

**Soundness:** 3 good
**Presentation:** 3 good
**Contribution:** 2 fair
**Rating:** 6
**Confidence:** 3

**Summary:**

This paper introduces the Gradient Subspace Distance (GSD), a metric to quantify the difference between two data sets: First, finding the gradient subspace of two data sets and then computing the distance between two subspaces.
The GSD was used in selecting public datasets in both pre-conditioning and transfer learning settings in this paper, and some experiments were done to support this.

**Strengths:**

1. The combination of public data and private is something interesting in differential privacy, and this paper follows that flow.
2 The. introduces the Gradient Subspace Distance (GSD) is something new in the measure of similarity of data sets.
3. The quality of presentation this paper is Good.

**Weaknesses:**

1. GSD-based public data set selection may leak sensitive information.

**Questions:**

1. What is running time (time complexity) of Algorithm 1 Gradient Subspace Distance (GSD) ?
2. It is unclear why Gradient Subspace Distance is a good measure of the similarity of data sets in nature.
3. In Lemma 4.1 what is the relationship between singular values and gradient subspace distance, which one is larger?

---

> ### Author Response · Authors · 2023-11-15
> **Thank you for your constructive feedback**
>
> We thank the reviewer for the constructive suggestions on improving the quality of this paper. Leveraging public data in private machine learning is an important direction to improve utility and we are encouraged that the reviewer also agrees that GSD is something new under this context.
>
> Below we give detailed responses to your concerns and questions.

---

> > ### Author Response · Authors · 2023-11-15
> > **Addresses weaknesses**
> >
> > We acknowledge that GSD is indeed not differentially private, and thus may leak sensitive information.  We give the following reasons to explain that 1) GSD is unlikely to breach privacy in practice, and 2) this work still makes meaningful contributions
> >
> > * The selection of a public dataset is a hyperparameter: in essentially all work on private ML, hyperparameter tuning is performed non-privately, since the privacy impact is considered to be minimal [1][2].
> > * Under this suggestion, in the context of public data sets selection, we assume what an adversary can know from GSD is “CIFAR-100 is the chosen public dataset for CIFAR-10”. Since this provides little information to the adversary, we also assume an unrealistic adversary that knows: 1) the model used by GSD and all related computations, such as gradients and hidden layer activation, and 2) 90% of the private data examples used by GSD, along with all the remaining private examples in the dataset. The adversary attempts to use membership inference attacks to determine if a particular image $x_i$ is used by GSD (if $x_i$ is in the remaining 10%). When we apply a well-studied and well-cited white-box membership inference attack, the experiments show that even under this overly powerful adversary, the attack success rate is as good as random guesses (see Appendix D.1 for more details).
> > * We believe the main contribution of this paper is: identifying this scientific question (exploring which public datasets perform well for a particular private task), and giving a simple yet effective solution.
> > * Nevertheless, we discuss DP methods for GSD computation in Appendix C. We hope that this work will serve as a starting point and inspire future research to effectively solve this scientific question with differential privacy.
> >
> > [1] Nicolas Papernot and Thomas Steinke. Hyperparameter tuning with renyi differential privacy. In Proceedings of the 10th International Conference on Learning Representations, ICLR ’22, 2022.
> >
> > [2] Shubhankar Mohapatra, Sajin Sasy, Xi He, Gautam Kamath, and Om Thakkar. The role of adaptive optimizers for honest private hyperparameter selection. In Proceedings of the Thirty-Sixth AAAI Conference on Artificial Intelligence, volume 36 of AAAI ’22, pp. 7806–7813, 2022.

---

> > ### Author Response · Authors · 2023-11-15
> > **Answer question2**
> >
> > The reviewer raises an interesting question: how will GSD reflect the similarity between data sets in nature? _GSD is designed to rank the utility of potential public data sets in the context of private machine learning._ Thus, it is natural to think that data sets that are semantically close, or say similar in nature, will have smaller GSD, meaning higher utility. However, our experiments show that this is not always true. We surprisingly find that CIFAR-100 is even ‘closer’ to SVHN w.r.t. gradient subspace distance, compared with MNIST-M (colorful MNIST). But in nature, MNIST-M should be closer to SVHN than CIFAR-100, as MNIST-M and SVHN are all images with digits. On the other hand, Task2Vec, the method we compare GSD with, is good at describing data set similarities in nature, but fails to give correct predictions for the utility w.r.t. private machine learning.
> >
> > That said, it is natural to think that smaller GSD means two data sets are more similar in nature. But this isn’t always true, and vice versa: two naturally similar data sets may not have smaller GSD. For example, CIFAR-10 should be most naturally similar to itself, but our experiments show that CIFAR-100 gives the smallest GSD. Again, the idea of GSD is to rank the utility of a list of potential public data sets in the context of private machine learning, and GSD is effective in doing so.

---

> > ### Author Response · Authors · 2023-11-15
> > **Answer question3**
> >
> > We appreciate the reviewer for proposing this interesting question. We are unsure if we understand the question, but we will attempt to provide an answer. _Empirically_, top singular values are usually larger. We have an empirical evaluation in Appendix D.2. Gradient subspace distance is bounded by $\sqrt{k}$, and is usually much smaller than top singular values. _Theoretically,_ this question can be extended to “does singular value itself have a mathematical relation with the rank of the matrix?” To the best of our knowledge, we cannot find a good mathematical relation between these two quantities.
> >
> > Please let us know if this answers your question. We are happy to discuss more.

---

> ### Author Response · Authors · 2023-11-15
> **Answer question1**
>
> One thing worth noticing about GSD is: it doesn’t update or iterate over the model, meaning the gradient subspace distance is given only by one pass through the model. _For a simple theoretical analysis_, we assume that the model used for GSD is a linear model with one hidden layer of size $n$, the number of examples are $m$, the input dimension is $d$, the output is of size $c$, and the lower dimension is $k$. Then the time complexity of GSD is $O(2mdn + 2mnc + 2m(dn+nc)\log(k) + 2(m+dn+nc)k^2)$ as we only need to get top-$k$ basis. Empirically, in our experiments, GSD computation takes only a _negligible_ amount of time. For example, giving a gradient subspace distance number in Table 1, like 0.15, will only take ~12s with one NVIDIA A100 (40GB) GPU (here #parameters is $O(10^4)$). For fine-tuning LoRA on ViT (#params becomes $O(10^5)$), the time cost is ~41s.

---

### Official Review · Reviewer_xFXu · 2023-11-01

**Soundness:** 4 excellent
**Presentation:** 4 excellent
**Contribution:** 3 good
**Rating:** 6
**Confidence:** 3

**Summary:**

The paper considers finding good representative subspaces for the gradients of loss functions when training ML models using sensitive data. In particular, these gradient subspaces are obtained by evaluating the gradients using public data. The background for this: DP-SGD introduces lot of noise and degrades the model performance as the parameter dimension grows. To this end, certain projection method have been introduced (e.g., Yu et al., 2021) where an orthogonal projector is used such that the DP-noise is added only in the projected space, reducing the total expected 2-norm of the injected DP-noise from $O(\sqrt{d})$ to $O(\sqrt{k})$, where $d$ is the parameter space dimension and $k$ the dimension of the projection subspace. Then, the problem is, how to obtain a good basis for the projector. Doing this privately is challenging, and a natural choice is to use public data for this. Then the question is, which public data set to use, so that the basis would well represent the subspace where the sensitive gradients live. This paper proposes a certain metric, "Projection Metric", to evaluate the goodness of the projector obtained with the public data. This metric is studied both theoretically and experimentally. Another related contribution is to consider "second phase pre-training", where a public data pre-trained large model is fine-tuned with another public data by having a small number of trainable parameters, and then the "Projection Metric" can be used to select best possible public dataset for this second phase pre-training, in case we use some projection method in the final fine-tuning with the sensitive data.

**Strengths:**

- Very well written paper, everything is explained clearly and comprehensively.

- Nice contributions with introducing the projection metric and studying its properties and also with the second-phase pre-training (as the authors point our, it has not been considered before).

- Extensive and clearly explained experiments.

**Weaknesses:**

- The biggest questions in my mind after reading the paper are related to the computational efficiency of the method. I think these questions are related to these projection methods in general, but of course are directly related to using this projection metric also. I don't really see it discussed anywhere, in the appendix either. Suppose I use that second phase pre-training such that I DP fine-tune LoRa parameters using some public dataset. There would be some $O(10^4)$ parameters, let's say there are 40k of them. And the public dataset size would be, let's say $O(10^5)$. Wouldn't computing the $V_{public}$ using SVD be quite expensive in this case? Or should I somehow limit the number of trainable parameters, the public dataset size, or use stochastic approximations to obtain $V_{public}$, or some other approximative numerical methods? As far as I see, one should update $V_{public}$ quite frequently? How frequently? I am just trying to think of a practical scenario, and what would one need to take into account when using these projection methods and this projection metric. E.g., when I compare public datasets, which one to use for construction the projector, should I just take some random subsets of them as candidates and would that be sufficient?

-  Overall, I think the presented ideas are intuitive and I believe useful but on theoretical level the contribution is not big, the value is on the experimental side. All in all this is a nice contribution and I appreciate also the "second phase fine-tuning" part of the paper and the careful experiments. I think this paper would fit well to this venue.

**Questions:**

- I have mostly questions related to the computational efficiency (see above). In the experiments of this paper, how big was the computational burden of choosing and using the projectors? I mean if you compare, e.g., to DP-SGD?

Comment: I think the form of the "projection metric" with those cosine principal angles as given in Definition 2 is quite intuitive, but I think the form where it is written with the Frobenius norm (used e.g. in the proof of Lemma 4.1) makes it even clearer, perhaps you could consider moving that to the main text? Just to quickly mention it.

Minor comments:

- Dot missing after Eq. 4
- There are some dots left to the tables all over, e.g. on page 8 and in the appendix.
- Page 20, paragraph "Experiment Setting", third line: bracket missing
- Page 21, before D.3 title: dot in the middle of the page

---

> ### Author Response · Authors · 2023-11-15
> **Thank you for the insightful and encouraging feedback**
>
> We appreciate the reviewer's insightful and encouraging feedback on the utility of public datasets in private machine learning. We are encouraged by the reviewer's interest in our idea and the acknowledgment of the paper's contribution, as well as its fit with ICLR. The summary given by the reviewer is thorough and concise, and we are lucky to have a reviewer who understands our work well.
>
> Below, we respond to your specific concerns.

---

> ### Author Response · Authors · 2023-11-15
> **Address weaknesses**
>
> The reviewer raised concerns about the computational efficiency of our method, which is an important aspect for all projection-based preconditioning methods. The short answer is, computational efficiency is not a problem for our methods. We give the following reasons:
> * We assume that the practitioner has a relatively small number of public data examples, which is consistent with projection-based pre-conditioning methods literatures[1][2]. This setting is especially applicable in certain key settings, such as a clinical environment where privacy and safety concerns make it inappropriate to use large, uncurated Internet datasets as public data due to the risk of data poisoning attacks. Instead, the practitioner may have a small set of trusted public datasets. In our experiments, we set the number of public data examples to 2000 or 300, depending on the size of the private dataset.
> * Another advantage of GSD is, GSD doesn’t update or iterate over the model, meaning the gradient subspace distance is given only by one-pass through the model. In Section 4.2, we empirically validate that the order of GSD is preserved over training, thus only two computation (one for $V_{pub}$ and one for $V_{priv}$) is needed to give the gradient subspace distance. With this advantage, combined with small number of public examples assumption, GSD computation only takes negligible time. For example, giving a gradient subspace distance number in Table 1, like 0.15, will only take ~12s with one NVIDIA A100 (40GB) GPU (here #parameters is $O(10^4)$). For fine-tuning LoRA (#params becomes $O(10^5)$), the time cost is ~41s.
> * For those projection-based methods, like GEP, yes, these methods will cost an extra computational burden. As pointed out by the reviewer, computing the basis can be quite expensive, and for GEP, $V_{pub}$ needs to be updated per iteration. However, extra computational burden makes GSD valuable as we can identify the best one from a list of public datasets in advance and thus, saving the time to run GEP over all the public datasets.
> * Practical scenario: as mentioned earlier, GSD is applicable in certain key settings, like clinical environments. To improve accuracy using private ML methods with public data assistance, practitioners can follow these steps: 1) Use GSD to rank the utility of available public datasets, and select the best one; 2) Use the same public examples in step 1, run algorithms like GEP to enhance accuracy.
>
> [1] Da Yu, Huishuai Zhang, Wei Chen, and Tie-Yan Liu. Do not let privacy overbill utility: Gradient embedding perturbation for private learning. In 9th International Conference on Learning Representations, ICLR 2021, Virtual Event, Austria, May 3-7, 2021. OpenReview.net, 2021. URL https://openreview.net/forum?id=7aogOj_VYO0.
>
> [2] Yingxue Zhou, Steven Wu, and Arindam Banerjee. Bypassing the ambient dimension: Private sgd with gradient subspace identification. In International Conference on Learning Representations, 2021. URL https://openreview.net/forum?id=7dpmlkBuJFC.

---

> ### Author Response · Authors · 2023-11-15
> **About comments**
>
> We thank the reviewer for pointing out Forbenius norm is more intuitive from an audience’s perspective. We also thank the reviewer for pointing out the minor mistakes made in this paper. We will correct these in the final camera-ready paper (if possible).

---

> ### Comment · Reviewer_xFXu · 2023-11-17
>
> Thank you for the replies. I would still have few questions about this:
>
> How big part of the subspace distance compute cost consists of computing the bases $V_{pub}$ and $V_{priv}$? I.e. if it takes 41s for LoRA, how many seconds of it is just computing $V_{pub}$ and $V_{priv}$? If number of public data samples is 2000, I would imagine it is quite expensive to compute just $V_{pub}$ using SVD.
>
> And a related question (concerns again these projection methods in general): How big part of the overall compute cost does forming $V_{pub}$ constitute in a common scenario? How does its cost compare to computing the clipped gradients?

---

> > ### Author Response · Authors · 2023-11-17
> > **About the time cost of computing subspace basis**
> >
> > Thank you for the detailed questions, they help us to understand deeply. Below, we respond to your specific question.
> >
> > **GSD**
> > * GSD only needs top-$k$ basis, i.e., let $p$ denote the model dimension, then $V_{pub}$ is $R^{p \times k}$. Here $k << p$, like if $p$ is $O(10^5)$, k is $O(10^1)$ or $O(10^2)$.
> > * Previous research has provided effective methods for computing the top-$k$ basis. Like, PyTorch has the built-in function `torch.svd_lowrank`, which uses randomize SVD to compute top-$k$ basis.  _For a simple theoretical analysis_, we assume that the model used for GSD is a linear model with one hidden layer of size $n$, the number of examples are $m$, the input dimension is $d$, the output is of size $c$, and the lower dimension is $k$. Then the time complexity of GSD is $O(2mdn + 2mnc + 2m(dn+nc)\log(k) + 2(m+dn+nc)k^2)$. Here, $O(mdn + mnc)$ is the time cost for computing the gradient matrix and $O(m(dn+nc)\log(k) + (m+dn+nc)k^2)$ is the time cost for computing $V_{pub}$. We can see that computing $V_{pub}$ takes a longer time than computing gradient matrix, but only scales by a small factor w.r.t $k$.
> > * In our implementation, considering the fact that GSD computation only requires negligible time anyhow, we turn to PyTorch Reduced SVD `torch.linalg.svd` for its better implementation. Here, the time complexity of computing $V_{pub}$ is $O(m(dn+nc)r)$, where $r$ is the rank of the gradient matrix. **Empirically**, we re-run the 41s LoRA experiments mentioned before. Computing $V_{pub}$ and $V_{priv}$ takes around 35s. This is slightly off from the previous analysis, as we are expecting this time cost to be a little bit shorter. We guess it is possibly due to PyTorch's SVD implementation not being as efficient as backprop on GPU.
> > * Additionally, GSD itself only requires negligible time, mainly because of its one-pass computation.
> >
> > **Projection methods**
> > * We borrow the complexity analysis from the GEP paper here. GEP uses power iteration to effectively compute top-$k$ basis. Let $k$ denote the lower dimension, $m$ be the number of public examples, $n$ be the size of private dataset and $p$ be the model dimension. The computation cost of one power iter: $O(2mkp + pk^2)$. The authors also use parameter grouping to reduce this computation cost to $O(2mkp/g + pk^2/g^2)$, where $g$ is the number of groups.
> > * For computing the clipped gradient, we borrow the analysis from this paper [BWZK22]. Using regular DP-SGD, computing a clipped gradient (one forward and one backprop) will take approximately $O(14mp)$. While computing the top-$k$ basis is less time-efficient than regular DP-SGD, the time cost won't increase significantly in practice since $k << p$.
> >
> > We again thank the reviewer for the follow-up question. It also helps us gain a better understanding of GSD and projection methods. The analysis provided is rough and may be incorrect. We are open to discussing this further with the reviewer.
> >
> > [BWZK22] Z. Bu, Y.-X. Wang, S. Zha, and G. Karypis, “Differentially private optimization on large model at small cost,” arXiv [cs.LG], 2022.

---

> > > ### Comment · Reviewer_xFXu · 2023-11-22
> > >
> > > Thank you for the reply! I didn't realize one can use also the randomized SVD for the task of computing $V_{pub}$, indeed it seems to become approximately the same order as the gradients computation. And that also seems to be the case with the projection method analysis / [BWZK22] gradient computation cost analysis. I believe that this analysis gives the correct idea. Thank you for the discussion.

---

### Official Review · Reviewer_Ukt9 · 2023-11-01

**Soundness:** 2 fair
**Presentation:** 2 fair
**Contribution:** 1 poor
**Rating:** 3
**Confidence:** 4

**Summary:**

This paper extends on the recent line of work that gradients during private optimization lie in a low-dimensional subspace and hence we can reduce the curse of dimensionality by leveraging this fact. Unfortunately, since estimating the subspace privately also incurs an error that scales with the dimension, these works estimate the subspace using "publicly" available dataset and use it as a proxy to project the gradient computation on private data to the low-dimensional subspace. This paper provides a metric that measures the distance between private and public subspace.

**Strengths:**

The definition of the metric.

**Weaknesses:**

The metric studied in the paper is studied a lot in low-rank approximation and non-private subspace estimation problems. In fact, the entire theory of Davis-Kahn revolves around such a metric. So, I really do not get the main contribution of the paper.

The bound on the reconstruction error is weird. On the right hand side you are measuring the error in terms of spectral norm while the bound is wrt the Frobenius norm. It is never desirable, starting the matrix approximation theory studied from early 20th century.

The proof idea in the paper has been used in several places and is not new at all. I would suggest the authors to do a proper survey of the literature in matrix analysis. Frank McSherry's thesis might be a good starting point to see the relevant literature from the CS perspective. If the authors want, I can suggest some literature from matrix perturbation theory and operator theory where these concepts are also widely studied. Stewart-Sun's book can be a good starting point.

**Questions:**

No question. I believe I understand the paper well.

---

> ### Author Response · Authors · 2023-11-15
> **Thank you for the feedback**
>
> We nevertheless appreciate the feedback from the reviewer, but we disagree with their understanding of the paper's main contribution. _The main contribution of this paper is not about the matrix approximation theory or the proof idea_. In fact, we don’t even acknowledge it as a contribution of this paper.
>
> The main contribution of this paper is proposing a crucial scientific question and providing a solution to it. All other reviewers, even the reviewer who recommended rejection can identify this contribution and agree with this assessment.
>
> We struggle to construct a rebuttal as the reviewer's feedback does not seem constructive or insightful. Nevertheless, we are open to suggestions about _matrix perturbation theory and operator theory_, and we are willing to explore how these early 20th-century studies can help enhance the quality of this paper.

---

> > ### Comment · Reviewer_Ukt9 · 2023-11-16
> > **Thanks for your response**
> >
> > Thanks for clarifying that your main contribution is not with respect to anything on matrix perturbation theory; however, the idea of using cosine similarity to measure the closeness of two subspaces is not new. It is a crucial scientific question that has been raised a long time ago. So, I am still unsure how this is anyhow a "new" question.
> >
> > Using Davis-Kahn type results is pretty common in understanding low-rank structure. As I mentioned earlier, there is a rich literature on it in various disciplines of applied mathematics. AFAIK, even in privacy, results like "Beyond worst case for singular value computation" and its follow-up work by Hardt-Price, and "Analyze Gauss" and its follow-up based on Dyson Brownian motion also use such similarity to study how close the span of the space is.
> >
> > Finally, I still do not understand how measuring bounding the spectral norm of error with Frobenius norm of related object is anyhow meaningful. If you want to prove a bound, you would like to impose similar metric. Spectral norm approximation is traditionally harder (even from the streaming literature) because you are trying to bound the error with respect to $\sigma_{k+1}$ instead of $\sum_{j \geq k} \sigma_{j}$. That is why results like Kapralov-Talwar and Achliptas-McSherry (in privacy) is beautiful and the space required for such an approximation even without privacy is linear in dimension. When you use spectral norm as a metric on one side and Frobenius norm on the other side, you end up comparing apples with oranges.

---

> > > ### Author Response · Authors · 2023-11-16
> > > **Thank the reviewer for the timely response**
> > >
> > > We thank the reviewer for the timely response. However, since this is the _second_ time the reviewer misunderstood the contribution of our paper, we decided to change our narrative. Let’s suppose given a scientific question: _exploring which public datasets perform well for a particular private task_. How will we approach this question? Using Frechlet inception distance? Training a binary classifier to access one example by one example? Prior to our work, _no_ research had addressed this question or offered a solution. In this work, we reduce this problem to “measure the closeness of two subspaces”, and demonstrate that this approach can effectively address the question.
> > >
> > > However, according to the reviewer's reasoning, it seems to us that we can never reduce this problem to “measure the closeness of two subspaces”, as this is “not new”. Indeed our solution incorporates previous research on the low-rank structure, but that is exactly the meaning of doing research, right? Building on previous studies is a common practice in research, and if existing research can effectively contribute to solving a new problem, there is _no_ need to reinvent the wheel just for the sake of novelty. According to the reviewer's reasoning, any paper that builds on pre-existing knowledge to address a new problem is considered "not new." For example, the Wasserstein GAN may be considered "not new" based on the reviewer's viewpoint, as the Wasserstein distance (also known as the earth mover distance) has been well-studied over the past decades. However, the impact of Wasserstein GAN in current times is undeniable.
> > >
> > > Lemma 4.1 in our paper may not provide a tight bound, but it still serves its purpose by proving that the utility of those public-data-assist private machine learning methods is bounded by GSD through Lemma 4.1. We welcome any constructive feedback to improve the analysis in our paper, _particularly suggestions on how to relate excess risk with GSD and conduct a tighter analysis_. However, based on the reviewer's feedback, it appears they are unable to offer constructive or insightful suggestions. We acknowledge that results like Kapralov-Talwar and Achliptas-McSherry (in privacy) is beautiful, and we also believe $e^{ix} = \cos{x} + i\sin{x}$ is beautiful. But they are inconsequential in this context. Using them as an analogy is not just comparing apple with orange, but comparing apple with hypokalemia.

---

### Meta-Review · Area_Chair_v3pJ · 2023-12-04

**Metareview:**

Summary: This paper attempts to find a good public data to use to find the best subspaces to reduce the dimension of gradients in DP-SGD. The core concept the authors use is "Projection Metric" which they use to evaluate the goodness of the projection (subspaces) obtained from public data. The authors study this projection metric both in theory and in practice.

Strength: the question the authors asks is timely and the presentation is well made in the paper. However,

Weaknesses: there is a disagreement between the authors and some of the reviewers regarding the upper-bound of L2-norm of the projector difference with its Frobenius norm. It would strengthen their paper if the authors clarify this issue in their future submission.

**Justification For Why Not Higher Score:**

The reviewers' concerns regarding the upperbound, whether or not it's sensible, seem a real concern.

**Justification For Why Not Lower Score:**

NA

---

### Decision · Program_Chairs · 2024-01-16

Reject